# PR-MCS: Perturbation Robust Metric for MultiLingual Image Captioning

**Yongil Kim[1]**     **Yerin Hwang[2]**     **Hyeongu Yun[1]**
**Seunghyun Yoon[3]**     **Trung Bui[3]**     **Kyomin Jung[1,2†]**

[1]Dept. of Electrical and Computer Engineering, Seoul National University
[2]IPAI, Seoul National University, [3]Adobe Research
{miles94, dpfls589, youaredead}@snu.ac.kr,
{syoon, bui}@adobe.com, kjung@snu.ac.kr

## Abstract

Vulnerability to lexical perturbation is a critical weakness of automatic evaluation metrics for image captioning. This paper proposes **P**erturbation **R**obust **M**ulti-Lingual **CLIPS**core(**PR-MCS**), which exhibits robustness to such perturbations, as a novel reference-free image captioning metric applicable to multiple languages. To achieve perturbation robustness, we fine-tune the text encoder of CLIP with our language-agnostic method to distinguish the perturbed text from the original text. To verify the robustness of PR-MCS, we introduce a new fine-grained evaluation dataset consisting of detailed captions, critical objects, and the relationships between the objects for $3,000$ images in five languages. In our experiments, PR-MCS significantly outperforms baseline metrics in capturing lexical noise of all various perturbation types in all five languages, while maintaining a strong correlation with human judgments.[1]

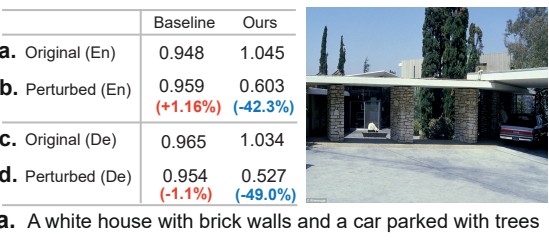

Figure 1: An example for perturbation robustness test. The baseline metric shows similar scores to both original and perturbed captions, but our metric shows prominent score drop for perturbed captions indicating that perturbation is well detected.

## 1 Introduction

Image captioning (Xu et al., 2015; Vinyals et al., 2015, 2016; Lu et al., 2017) is a multimodal task that automatically generates captions that describe the visual content of an image and integrates multiple disciplines of visual and textual modality. Image captioning is a natural language generation (NLG) task (Gatt and Krahmer, 2018), but the evaluation metric has different characteristics from other NLG metrics (Sai et al., 2022). Image captioning metrics should be able to evaluate not only linguistic fluency and syntactic thoroughness but also semantic correspondence to visual content(Bai and An, 2018).

Evaluation criteria for image captioning have evolved from N-gram-based metrics (Papineni et al., 2002; Lin, 2004; Banerjee and Lavie, 2005; Vedantam et al., 2015) to reference-free metrics (Lee et al., 2020, 2021; Hessel et al., 2021). Recently, CLIPScore (Hessel et al., 2021) has been proposed to leverage the large-scale pretrained vision and language model CLIP (Radford et al., 2021). In evaluating generated captions by computing cosine similarity between embedded vectors (i.e., image and text) using CLIP, CLIPScore achieves a higher correlation with human judgments than traditional metrics.

However, Sai et al. (2021) have revealed that current metrics are prone to failure in capturing lexical noise in generated captions. For example, when a perturbation is applied to an original caption (e.g., a removal or swap at the token level), existing image captioning metrics do not recognize the change and compute a score similar to that for the original caption case. This failure to capture lexical noise raises a critical question concerning the reliability

---

[†]Corresponding author.

[1]All the datasets and code are available in https://github.com/yong1-kim/PR-MCS. Our dataset is distributed under the CC-BY-NC 4.0 license.

of the metric, as shown in the example in Figure 1. CLIPScore exhibits the same tendency in our analysis, which reflects its vulnerability to perturbed texts. By extending CLIPScore to a multilingual setting, we observe that a multilingual CLIPScore exhibits the same limitations in multiple languages other than English, i.e., French, German, Spanish, and Japanese.

In this paper, we address this problem by proposing a novel method for enhancing the perturbation robustness of CLIPScore. Our method is to fine-tune the text encoder of CLIP with perturbed captions so that the text encoder can distinguish the perturbed text embeddings from the original text embeddings. The simplicity and effectiveness of our method enable us to apply it to multiple languages without relying on human annotations. Using our method, we develop Perturbation-Robust Multilingual CLIPScore (PR-MCS), a perturbation-robust and language-agnostic metric for image captioning.

Furthermore, to validate the robustness of PR-MCS against perturbations and its high human correlation, we introduce two newly created datasets: M-FineCap3k and M-CapEval1k. Currently, most image captioning datasets are limited to English, necessitating a machine translation (Bahdanau et al., 2014; Johnson et al., 2017) process for multilingual experiments. However, this process relies on the performance of the translation model, which may result in lower evaluation reliability compared to human-annotated labels. Hence, we elicit image captions directly from human experts, tailored to the purpose of the datasets. Firstly, M-FineCap3k is designed as an image captioning dataset, created to generate fine-grained captions that are appropriate for the corresponding images. Secondly, M-CapEval1k serves as a benchmark dataset developed for the purpose of measuring the human correlation with image captioning metrics.

Finally, experimental results on five datasets, including M-FineCap3k, demonstrate that PR-MCS outperforms baseline metrics in capturing lexical noise in captions across all five languages considered. In addition, the results of measuring human correlation using M-CapEval1k reveal that PR-MCS exhibits a strong alignment with human judgments. Therefore, we confirm that the proposed PR-MCS is a useful and reliable image captioning metric with perturbation robustness and a strong correlation with human judgments.

## 2 Related works

**Image captioning metrics** As with other natural language generation tasks, image captioning can be evaluated using various proposed metrics. BLEU (Papineni et al., 2002), ROUGE (Lin, 2004), and METEOR (Banerjee and Lavie, 2005) are representative image captioning metrics based on n-gram similarity with reference captions. Other widely used reference-based metrics include CIDEr (Vedantam et al., 2015), which weights n-gram similarity (Kondrak, 2005) through TF-IDF (Aizawa, 2003), and SPICE (Anderson et al., 2016), which evaluates captioning based on scene graphs. Recently, reference-based metrics using embedding similarity with reference captions based on a model, such as BERTScore (Zhang et al., 2019), BERT-TBR (Yi et al., 2020), and VilBERTScore (Lee et al., 2020), have been introduced.

**CLIPScore** Researchers have also proposed unreferenced image captioning metrics that evaluate generated captions by comparing them with original images that does not require ground-truth caption (Madhyastha et al., 2019; Kusner et al., 2015; Lee et al., 2021; Chen et al., 2020). CLIPScore (Hessel et al., 2021), which is also a reference-free metric, relies heavily on the CLIP (Radford et al., 2021) model, trained with 400 million image caption pairs using a contrastive objective function that distinguishes original image–caption pairs from unmatched captions. The calculated CLIPScore is the weighted value of cosine similarity between image embedding and text embedding encoded by the CLIP model. Despite the fact that CLIPScore exhibits a high correlation with human evaluation, it is limited in that it is an image captioning metric that applies only to English. In this study, we propose a new multilingual image captioning metric developed by extending CLIPScore to a multilingual setting.

**Perturbation Robustness** In a recent study, Sai et al. (2021) selected various criteria for use in assessing how various NLG evaluation metrics perform. In addition, perturbation was applied to multiple image captioning factors to assess the perturbation robustness of the image captioning metrics. Sai et al. (2021) provided a perturbation checklist of metrics for NLG tasks; we go further and present a novel metric that overcomes the limitations of

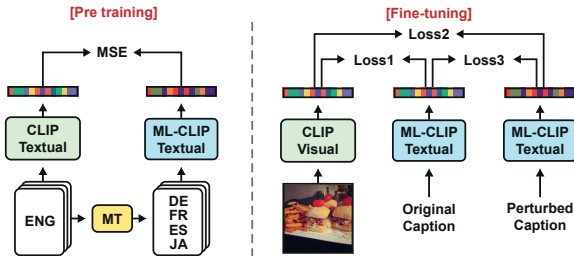

Figure 2: Overall training procedure of PR-MCS. We pre-train Multilingual text encoder with teacher learning. Then, we fine-tune the Multilingual text encoder.

other metrics. We select some perturbation criteria from among those suggested by Sai et al. (2021), designate them as target perturbations, and show that the CLIPScore cannot detect these perturbations in multiple languages. Even if the generated captions are corrupted, CLIPScore outputs similar results for the original and corrupted sentences. This study proposes a novel metric with perturbation robustness based on CLIPScore to address its weaknesses in multiple languages.

## 3 Perturbation Robust Multi-Lingual CLIPScore

### 3.1 MultiLingual CLIPscore

We propose a new multilingual image captioning metric, **M**ultilingual **CLIPS**core (MCS), to overcome the limitation of CLIPScore that it can only be applied to English. Recently, Fredrik Carlsson (2022) proposed a multilingual CLIP model applicable to various languages by learning the expressive power of CLIP's text encoder through teacher learning. Similarly, in this study, we pre-train the multilingual text encoder with teacher learning (Figure 2 left). In teacher learning, the multilingual text encoder learns the pre-trained CLIP text embedding of an English sentence so that the embeddings of the sentence translated through the machine translation model are similar. We use MSE loss as the teacher learning loss, and the formula is as follows:

$$L(t, s) = \frac{1}{N} \sum_{i=0}^{n} (t - s)^2,$$

where $t$ is the teacher embedding and $s$ is the student embedding. More details on multilingual textual encoder pre-training can be found in the appendix A.

We present a new multilingual image captioning metric, MCS, using this model as a backbone. MCS

| Metric \ Language | EN | DE | ES | FR | JA |
|---|---|---|---|---|---|
| CLIPScore | 0.270 | - | - | - | - |
| †MCS | 0.264 | 0.264 | 0.251 | 0.235 | 0.250 |
| *MCS | 0.268 | **0.265** | **0.253** | **0.236** | **0.252** |

Table 1: Human Correlations with Multi-CapEval1k. The † symbol denotes MCS metric with Fredrik Carlsson (2022), and the * denotes an MCS metric with our pretrained text encoder.

uses image–caption pairs with weight given to the cosine similarity of embeddings created through visual and text encoders, respectively. The formula for an image–caption input pair $(I, c)$ in MCS is as follows:

$$MCS(I, c) = w * max(0, cos(V(I), T(c)),$$

where $V(I)$ is the visual embedding where the image is passed through the visual encoder and $T(c)$ is the text embedding where the caption is passed through the multilingual text encoder. The value of $w$ is set to 2.5 as in the original CLIPScore.

### 3.2 Human Correlations of MCS

An adequate evaluation of image captioning metrics requires assessing the correlation between metric-generated caption scores and human-generated caption scores. However, to the best of our knowledge, no benchmark exists to evaluate image captioning metrics across multiple languages. For languages other than English, machine translation is necessary, but this approach can lead to incorrect results when machine translation models attempt to correct sentences that have already been annotated with a low score. To tackle this problem, we created the M-CapEval1k benchmark by translating CapEval1k (Lee et al., 2021), an image captioning metric evaluation set originally in English, into five languages (English, German, Spanish, French, and Japanese) with the assistance of native speakers for each language. Our translation process ensured that the goodness or badness scores for each sentence were maintained. This benchmark can be leveraged for the quantitative evaluation of multilingual image captioning metrics, and an example of the M-CapEval1k is provided in the Appendix C.

Table 1 shows the Kendall tau-c ($\tau_c$) value (Kendall, 1938) representing the human correlation for each language of the metrics. The MCS which uses our pretrained model as a backbone

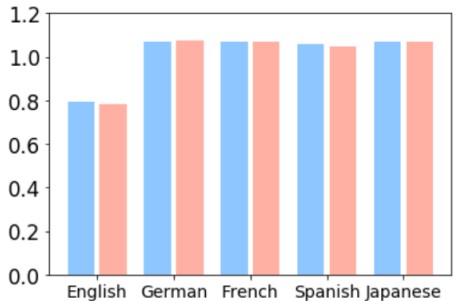

Figure 3: Scores of original captions (blue) and perturbed captions (red) with repetitive lexical noises. In all cases, the perturbed captions show no differences from the original captions.

demonstrates a high correlation with human judgement similar to CLIPScore in English. For other languages, it can serve as a baseline for multilingual image captioning metrics in future research. Therefore, the MCS presented in this paper extends CLIPScore well to languages beyond English.

### 3.3 Vulnerability to lexical perturbation

We employed some of the perturbation criteria identified by Sai et al. (2021) and checked the perturbation sensitivity of CLIPScore and MCS. One of the criteria, *"Repetition"*, is a perturbation in which words appear repeatedly at the token level in the original caption (e.g., "*I am a boy.*" → "*I am am a boy boy.*"). Figure 3 shows what score is given by baseline metrics when repetitive lexical noise is introduced. We randomly selected 3,000 samples from the MSCOCO (Lin et al., 2014) dataset, translated them into four languages, and gave repetitive lexical noise to the captions. The blue bars are the score for the original captions, and the red bars are the score for the perturbed captions. For English, CLIPScore is used as a metric, and for other languages, MCS is used for score extraction. The caption to which lexical noise is added is expected to have a lower matching score with the image than the original caption. However, for all languages, the scores for the perturbed captions are not lower than those for the original captions. There are even cases in which the perturbed caption is given a higher score. Similar tendencies can be observed for other perturbation criteria as well as *"Repetition"*. These results confirm that CLIPScore and MCS are limited in that they are vulnerable to lexical perturbation and that a metric that is robust to perturbation is needed.

### 3.4 Perturbation-Robust Multilingual CLIPScore (PR-MCS)

We introduce a novel language-agnostic perturbation method that increases the robustness of MCS. This method of fine-tuning the multilingual text encoder is to add three losses to original CLIP loss. The CLIPScore is constructed through embeddings of pre-trained CLIP without additional training. The CLIP loss $\mathcal{L}_{CLIP}$ consists of in-batch contrastive loss using cross-entropy loss, and the implementation is the same as the pre-training loss of CLIP. We construct a loss based on the contrastive loss of CLIP to maintain the high correlation with the human judgment of CLIPScore.

Then, we train the text encoder by adding three additional losses for perturbation robustness. These losses aim to maintain the close relationship between the image embedding and the original caption while increasing the distance from the perturbed caption. An (image, original caption, perturbed caption) triplet is then used as input to fine-tune the text encoder through three losses, as shown in Figure 2 (right). The losses are as follows:

$$\mathcal{L}_1 = 1 - cos(V(I), T(o)), \quad (1)$$
$$\mathcal{L}_2 = max(m, cos(V(I), T(p)), \quad (2)$$
$$\mathcal{L}_3 = max(m, cos(T(o), T(p)), \quad (3)$$

where $(I, o, p)$ is the (image, original caption, perturbed caption) triplet, $V$ is a visual encoder, and $T$ is a text encoder.

Equation (1) is composed of the cosine embedding loss of the two representations needed to increase the similarity between the image embedding and the original caption. Since MCS is based on cosine similarity, the purpose of Eq. (1) is to obtain a higher score for the original caption. Equation (2) reduces the marginal cosine similarity of image embedding and perturbed caption embedding. Equation (3) reduces the similarity between the original and perturbed caption embeddings. The margin $m$ is set to 0.1. These three losses are combined to obtain the final objective function as follows:

$$\mathcal{L} = \mathcal{L}_{CLIP} + \lambda_1 * \mathcal{L}_1 + \lambda_2 * \mathcal{L}_2 + \lambda_3 * \mathcal{L}_3.$$

We develop PR-MCS by fine-tuning Multilingual CLIP using the proposed loss function.

$$PR\text{-}MCS(I, c) = w * max(0, cos(V(I), T^*(c)),$$

where $T^*(c)$ is the text embedding from the fine-tuned multi-lingual text encoder. $w$ is also set to 2.5, as in the original CLIPScore and MCS.

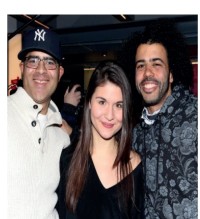

**Original Caption**
Deux hommes et une femme souriant surun fond noir
Two men and a woman smiling on a black background

**Critical Objects**  deux hommes      femme
                     two men          woman

**Relationship**     souriant
                     smiling

**Background**       fond noir
                     black background

Figure 4: An example of the proposed M-FineCap3k dataset (translation provided for explanation purpose).

## 4 M-FineCap3k Dataset

To evaluate the performance of PR-MCS, we introduce a new image captioning dataset, M-FineCap3k. Most existing image captioning datasets are limited to English (Young et al., 2014; Krishna et al., 2017). Therefore, a model for machine translation (MT) from English to other languages is essential to evaluate image captioning in various languages. However, translated evaluation set is highly dependent on the performance of the MT model and is highly likely to have an English-language bias. In addition, translation is inarticulate in the target language because it is difficult to reflect the unique characteristics of each language, such as word order and lexical choice (Zhang and Toral, 2019; Cao et al., 2020). Therefore, the results obtained using a translated evaluation set achieve poorer agreement with human evaluation than those obtained using the English evaluation set. For these reasons, a human-annotated image captioning evaluation set with a wide variety of languages is needed.

**Multilingual image captioning evaluation set**
We introduce a new human-annotated multilingual evaluation dataset, M-FineCap3k. We extended FineCapEval (Cho et al., 2022), which has only English captions, to five languages (English, German, French, Spanish, and Japanese). Human experts viewed the images for each language and added captions directly. Each sentence generated directly by native speakers is more fluent than translated versions. Moreover, M-FineCap3k can capture various cultural aspects that MT models cannot (Liu et al., 2021). Therefore, the reliability of evaluation in multilingual settings increases. An example of the dataset is shown in Figure 4. Human annotators for each language created a caption, critical objects, backgrounds, and relationships for a given image.

| Critical Objects | white shirt, grey shorts, golf, green field |
|---|---|
| Original | A man, wearing a white shirt and grey shorts, is playing golf on a green field with green trees and a blue sky in the background |
| Jumble | with green playing blue trees a background. green in shorts, and white the is wearing man, a A and grey on sky a golf shirt field |
| Removal | man, white and shorts, is playing a green trees a |
| Repetition | A man, man, wearing a white white shirt and and grey shorts, shorts, is is playing playing golf on a a green field with green green trees and a a blue sky in the background. |
| Masking | A [MASK] wearing a [MASK] shirt [MASK] grey [MASK] [MASK] [MASK] golf on [MASK] green field with [MASK] [MASK] and [MASK] blue sky in the background. |
| Substitution | A man, wearing a golf and green field, is playing white shirt on a grey shorts with green trees and a blue sky in the background. |

Figure 5: Example of perturbed captions of M-FineCap3k in English. The critical objects are shuffled for in-sentence substitution.

**Fine-grained caption with critical objects**
The widely-used multi-lingual image captioning dataset, Multi30K(Elliott et al., 2016, 2017), is also based on human annotations of images from Flickr30K(Young et al., 2014). However, the dataset is composed of brief sentences, rendering it challenging to execute diverse lexical perturbations. Moreover, it is only expanded to two languages, namely German and French. To address this, we create M-FineCap3k with long, detailed captions of 20 words or more to enhance the impact of human annotation and generate a range of perturbed captions for evaluation purposes. In addition, we had human experts point to critical objects, backgrounds, and relationships to create perturbed captions effectively. As described above, the image captioning metric should also reflect the semantic correspondence of whether the caption captures information contained in visual content well. The critical object of the caption should point to the most important object of this visual content, so it plays a key role in the comparison between embeddings. When perturbation is applied to this critical object, a more powerful and effective perturbation is achieved.

For instance, the well-known weakness of CLIP is that it does not produce different results when the positions of critical objects in the sentence are changed. For example, the CLIP text embeddings of the two sentences "*A **blue** car in*

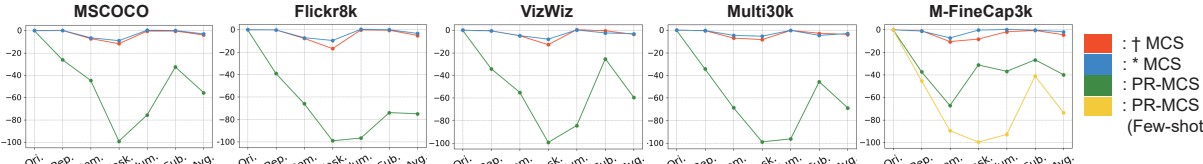

Figure 6: Experiment result graph. The y-axis value represents the score drop of the perturbed caption as a percentage difference compared to the original caption. The experiment is conducted with five datasets, and we report the average of the languages to confirm the results for each perturbation. As a result, it can be seen that PR-MCS is more robust than baseline metrics for all perturbations across all datasets.

*front of the **white** church*" and "*A **white** car in front of the **blue** church*" are almost identical. To evaluate the robustness to this perturbation, we construct perturbation criteria using critical object information. The statistics of the dataset can be found in the Appendix D.

## 5 Experiments

Our framework seeks to identify whether a given metric can detect lexical noise in a generated caption. Through exhaustive experiments, we evaluate whether the PR-MCS developed as described in this paper successfully distinguishes the perturbed caption from the original caption.

### 5.1 Experimental Setup

**Fine-tuning set** We use MSCOCO, the dataset most widely used for image captioning, as the fine-tuning set to enhance the perturbation robustness of MCS. We use the training and validation split of the MSCOCO dataset described by Chen et al. (2020). The number of elements in the training set is 414k. Since only English captions exist in MSCOCO, captions are translated into four other languages using the MBART-50 (Liu et al., 2020) MT model.

**Evaluation set** To comprehensively evaluate the perturbation robustness of PR-MCS, we choose four evaluation sets: MSCOCO, VizWiz (Gurari et al., 2020), Flickr8k (Anitha Kumari et al., 2019), Multi30K, and M-FineCap3k. As in the fine-tuning, MSCOCO, VizWiz, and Flickr8k, which have only English captions, are translated using the MBART-50 MT model.

**Baseline metric** As the baseline of the experiment, we use two MCS metrics. As mentioned above, the MCS metric is configured using CLIP's visual

and multilingual text encoder. The first baseline is the MCS metric constructed using the multilingual CLIP text encoder implemented by Fredrik Carlsson (2022) as the backbone. The second baseline is the MCS metric constructed using the multilingual text encoder trained by the teacher learning method described in Section 3.1.

### 5.2 Perturbation configuration

We select the following five criteria to perturb the sentences in the fine-tuning and evaluation sets. The criteria below are error types commonly found in model-generated captions. These criteria are part of the checklist proposed by Sai et al. (2021). We select five orthogonal criteria. Each perturbation example is shown in Figure 5.

**Repetition** Repeated words are found in several model-generated captions. A well-known problem is that the transformer model is vulnerable because it does not capture repetitive perturbation well at the embedding level. We give each word token a repeating perturbation with a probability of 0.4.

**Removal** Among the sentences given a low score in the evaluation dataset for the image captioning metric, such as Composite (Aditya et al., 2015) or Pascal50s (Vedantam et al., 2015), some word tokens are removed, and incomplete sentences are found. We configure perturbation by removing some tokens to reflect this noise. Each word token is drawn with a probability of 0.4.

**Masking** Masking is a perturbation in which randomly selected tokens in the caption are replaced with [Mask] tokens. When lexical noise is given in units of tokens, the meaning of the corresponding token disappears, but unlike in the Removal case, the position is maintained. Position information can be critical in a reference-free metric based on a transformer model such as CLIPScore (Dai

| Eval Dataset | Metric | EN | | DE | | FR | | ES | | JA | |
|---|---|---|---|---|---|---|---|---|---|---|---|
| | | Original | Perturbed Average | Original | Perturbed Average | Original | Perturbed Average | Original | Perturbed Average | Original | Perturbed Average |
| **MSCOCO** | MCS | 1.0939 | 1.0581 (-3.27%) | 1.0839 | 1.0551 (-2.66%) | 1.0837 | 1.053 (-2.83%) | 1.0799 | 1.0499 (-2.78%) | 1.0286 | 0.9813 (-4.60%) |
| | PR-MCS | 1.4177 | **0.2996** (-78.66%) | 1.3153 | **0.5156** (-60.80%) | 1.4127 | **0.6643** (-52.97%) | 1.4086 | **0.6814** (-51.63%) | 1.3948 | **0.9202** (-34.02%) |
| **Flickr8k** | MCS | 1.0915 | 1.0543 (-3.41%) | 1.0779 | 1.0455 (-3.00%) | 1.0758 | 1.0438 (-2.98%) | 1.0747 | 1.0422 (-3.03%) | 1.0293 | 0.9879 (-4.02%) |
| | PR-MCS | 1.1484 | **0.2402** (-79.09%) | 1.1621 | **0.2674** (-76.99%) | 1.6681 | **0.4275** (-74.37%) | 1.6534 | **0.4685** (-71.66%) | 1.6279 | **0.4463** (-72.58%) |
| **VizWiz** | MCS | 1.0626 | 1.0338 (-2.71%) | 1.0491 | 1.01 (-3.72%) | 1.0509 | 1.0251 (-2.46%) | 1.0505 | 1.0146 (-3.41%) | 1.0414 | 1.0058 (-3.42%) |
| | PR-MCS | 0.9769 | **0.2944** (-69.86%) | 0.9895 | **0.415** (-58.06%) | 1.442 | **0.6163** (-57.26%) | 1.4526 | 0.6257 (-56.93%) | 1.4113 | **0.6224** (-55.90%) |
| **Multi30k** | MCS | 1.0611 | 1.0148 (-4.36%) | 1.0625 | 1.0264 (-3.39%) | 1.0624 | 1.0247 (-3.54%) | - | - | - | - |
| | PR-MCS | 1.1008 | **0.3041** (-72.37%) | 1.096 | **0.3613** (-67.03%) | 1.5767 | **0.5114** (-67.56%) | - | - | - | - |
| **M-FineCap3k** | MCS | 1.0657 | 1.0372 (-2.67%) | 1.0803 | 1.047 (-3.09%) | 1.0575 | 1.0295 (-2.65%) | 1.0577 | 1.0297 (-2.65%) | 0.963 | 0.9856 (2.34%) |
| | PR-MCS | 1.0429 | **0.5579** (-46.51%) | 0.7261 | **0.4611** (-36.50%) | 1.3634 | **0.8488** (-37.74%) | 1.3821 | **0.9426** (-31.80%) | 1.1551 | **0.6133** (-46.91%) |
| | PR-MCS (with few shot setting) | 0.7125 | **0.1584** (-77.77%) | 0.589 | **0.1742** (-70.42%) | 1.4862 | **0.4391** (-70.45%) | 1.4292 | **0.3964** (-72.26%) | 1.4136 | **0.314** (-77.79%) |

Table 2: Experiement results table. Each values are represented using the average value for each perturbation. For all five datasets, PR-MCS outperforms the baseline performance for all languages, and the performance has further increased after additional M-FineCap3k fine-tuning in few-shot settings.

et al., 2019; Devlin et al., 2018; Ramachandran et al., 2019; Wu et al., 2021). Therefore, even if the [Mask] token does not appear in the generated caption, we select Masking perturbation as the criterion, separate from Removal, to address the above case. Each word token is replaced with a [Mask] token with a probability of 0.4.

**Jumble** We generate perturbed samples using random-order permutation at the token level in the original reference caption. The model composing the metric can see all tokens of the sentence, including visual content, but considerable noise is introduced into the position information.

**Substitution** Substitution involves changing the positions of key elements in a sentence. In the case of M-FineCap3k, substitution is performed using critical objects annotated by human experts. In the remaining datasets, nouns in the caption are extracted, and their positions are changed. The perturbed caption includes all elements that exist in the original caption, but unlike in the Jumble case, it does not deform the grammatical structure at all. Detecting substitution noise well is the most challenging task because it requires judging semantic correspondence to visual content perfectly.

## 5.3 Perturbation robustness evaluation

We report the main results for all datasets and languages in Figure 6 and Table 2. The robust evaluation metric is expected to give lower scores to perturbed captions than to original captions.

Each graph of Figure 6 shows the experimental results for MSCOCO, VizWiz, Flickr8k, Multi30K, and M-FineCapEval by perturbation. Each point represents the average results for five languages for one perturbation. It shows how much score drop the perturbed caption has from the original caption. The green line indicates PR-MCS, and the blue and red lines refer to the two baseline multilingual CLIPScores. The scores of the perturbed caption by baseline metrics do not differ much from those of the original caption for any perturbation methods. In some cases, the scores for the perturbed captions are higher than those for the original captions.

However, our metric exhibits a significant score decrease for all perturbations compared to the original captions, which means that the metric can clearly distinguish when the perturbation is applied. In other words, our metric exhibits robustness for all perturbations in the evaluation dataset. In particular, even in the cases of Repetition and Substitution, which are known to be challenging pertur-

bations, PR-MCS detects perturbations very well, while baseline metrics do not capture perturbations at all.

Table 2 shows the score of the original caption and the average score of the perturbed caption given by the metrics for each of the four evaluation sets for each language. The result shows how much the percentage of the score decreased for each perturbation in comparison to the original caption. The results for the baseline metrics show that the score decrease for the average perturbation is very small, i.e., approximately 3%, relative to the original caption. It is difficult to say that the metric can distinguish the perturbed caption from the original caption based on such a slight difference. In contrast, in the case of PR-MCS, the percentage decrease for the perturbed caption ranges from 50% to 70%. Clearly, our proposed method exhibits perturbation robustness in the metric score and can identify perturbed captions through anomaly detection only with a performance drop. In the cases of Vizwiz, Flickr8k, Multi30K, and M-FineCap3k, the performance is outstanding even though the distributions are not trained in fine-tuning.

## 5.4 Few-shot setting for M-FineCap3k

As the results summarized in section 5.3 show, PR-MCS is much more robust to perturbation than the baselines in M-FineCap3k. However, the performance degradation for perturbed captions in M-FineCap3k is lower than for MSCOCO, VizWiz, Flickr8k, and Multi30K (e.g., -55.66% to -39.89% in average from MSCOCO). We attribute this to the distribution shift from the sequence length difference between the MSCOCO fine-tuning set and the M-FineCap3k test set. VizWiz, Flickr8k, and Multi30K are composed of short captions, so there is not much difference in caption length between them and MSCOCO. Therefore, to check whether the distribution can be learned when some information about the M-FineCap3k 3K test set is provided, we perform additional experiments on M-FineCap3k with a few-shot setting. We split M-FineCap3k into subsets proportioned 1:9 in size and use only 300 perturbed captions as the few-shot input.

The experimental results for the few-shot setting are shown in Table 2 and Figure 6 (the yellow line in rightmost graph). When the distribution for the fine-grained caption is given, the overall performance in perturbation detection, in terms of the av-

|  | English | | Other Langs | | Multi- | Perturbation |
|---|---|---|---|---|---|---|
|  | $\tau_c$ | $\rho$ | $\tau_c$ | $\rho$ | Lingual | Robustness |
| **CLIPScore** | 0.270 | 0.408 | - | - | ✗ | ✗ |
| **MCS** | 0.268 | 0.401 | 0.250 | 0.385 | ✓ | ✗ |
| **PR-MCS** | 0.267 | **0.425** | **0.250** | 0.376 | ✓ | ✓ |

Table 3: Correlations with human judgment for M-CapEval1k.

| Dataset
Metric | Composite (EN) | Flickr8k (EN) |
|---|---|---|
| **BLEU-1** | 0.313 | 0.323 |
| **BLEU-4** | 0.306 | 0.308 |
| **ROUGE-L** | 0.324 | 0.323 |
| **METEOR** | 0.389 | 0.418 |
| **CIDEr** | 0.377 | 0.439 |
| **SPICE** | 0.403 | 0.449 |
| **PR-MCS** | **0.493** | **0.527** |
| **CLIPSCore** | **0.497** | **0.530** |

Table 4: Kendall Tau ($\tau_c$) correlations with human judgment for existing English datasets.

erage score, increases for all five languages. These results show that lexical noise in long sentences is more reliably captured by learning a small number of samples with a few-shot setting. The experimental results for all languages and all perturbations for each dataset are provided in the Appendix H.

## 5.5 Correlations with human judgement

Table 3 and Table 4 shows that PR-MCS is an useful image captioning metric with high correlation with human judgment. The higher the Kendall tau-c ($\tau_c$) value and the Pearson correlation coefficient ($\rho$) (Benesty et al., 2009), indicators for viewing the correlation with human judgment, the better. The Kendall tau-c value is the similarity between the two variables based on ranking, and the Pearson correlation coefficient is a measure of linear correlation between two sets of data.

For Table 3, M-CapEval1k is used as an evaluation set for measuring the performance of image captioning metric, and it demonstrates excellent characteristics of PR-MCS for various languages within the dataset. As introduced in Section 3.2, our created M-CapEval1k serves as a benchmark for measuring the correlation of given metrics with human judgment across various languages. PR-MCS exhibits perturbation robustness while also showing similar performance to CLIPScore and MCS with Kendall correlation in English and even higher performance in Pearson correlation.

In Table 4, it is evident that PR-MCS remains

a valuable metric for existing English benchmark datasets such as Composite (Aditya et al., 2015) and Flickr8k_Expert (Hodosh et al., 2013). The correlation of PR-MCS with human judgment for English sentences is notably strong compared to conventional reference-based metrics and is similar to CLIPScore. Furthermore, as asserted in Table 3, PR-MCS operates in a multilingual setting and demonstrates perturbation robustness, distinguishing it from CLIPScore. PR-MCS's adaptability to a multilingual context allows for showcasing its effectiveness by translating these datasets into languages besides English using machine translation. PR-MCS consistently demonstrates strong human correlation across all other languages, and the comparative results with reference-based image captioning metrics can be found in Appendix I.

In light of the cumulative findings discussed previously, for sentences that potentially include perturbations that an image captioning model may output, existing metrics such as CLIPScore and MCS are vulnerable. PR-MCS is a useful metric that can evaluate good sentences positively and bad sentences negatively for potentially perturbed sentences; thus, it can be used instead of CLIPScore.

## 6 Conclusion

In this paper, we propose PR-MCS, a perturbation-robust metric for multilingual image captioning using language-agnostic fine-tuning. PR-MCS, developed by fine-tuning the text encoder of CLIP, can distinguish lexically perturbed text from original text. We also propose a new fine-grained multilingual image captioning evaluation set, M-FineCap3k, for use in perturbation robustness evaluation. Experimental results for existing datasets and our new dataset show that PR-MCS detects perturbation well and is robust to perturbation in multiple languages. Furthermore, we verify that PR-MCS is a useful metric with strong correlation with human judgment, using our M-CapEval1k.

## Limitations

### Model bias of machine translation in training
In our study, an evaluation set is created by directly annotating languages other than English to remove the bias of the machine translation (MT) model in the evaluation phase. However, in the training phase, the dataset size is too large to annotate directly in multiple languages other than English. Therefore, the pre-training set and the fine-tuning

set are translated into other languages by utilizing the MT model, so we have no choice but to depend on the performance of the MT model and avoid model bias.

Recently, Reimers and Gurevych (2019) released the MultiLingual CLIP based on the ViT-B/32 CLIP model*. We constructed an MCS with this model as the backbone and measured human correlation using M-CapEval1k. The results showed a slightly lower correlation compared to the MCS with our implemented Multilingual Text Encoder, but were still similar (average Kendall score for 5 languages: Sentenceformer based MCS - 0.249, our MCS - 0.255). However, as this model is a large-scale model, it has the disadvantage of being slow in inference speed as an automatic metric. We conducted finetuning with our more lightweight model with high human correlation. In future work, if experiments are conducted with this large-scale model, additional analysis on our proposed methodology and new metrics can be done through a variety of experimental results and interpretations. Furthermore, trends in large-scale models can also be confirmed.

## Ethics Statement

The annotators for the two newly created datasets (M-FineCap3k and M-CapEval1k) were hired through a data annotation service. The remuneration was calculated according to the annotators' country of residence, as determined by the company. Annotators were asked not to write any toxic content (1. offensive, sexist, or racist comments; 2. toxic words; 3. sexual behavior). All other datasets and models used in the experiments are from the publicly available website or Github.

## Acknowledgements

We thank anonymous reviewers for their constructive and insightful comments. K. Jung is with ASRI, Seoul National University, Korea. This work was supported by Samsung Electronics and the BK21 FOUR program of the Education and Research Program for Future ICT Pioneers, Seoul National University in 2023. This work was partly supported by Institute of Information & communications Technology Planning & Evaluation (IITP) grant funded by the Korea government(MSIT)

---

*https://huggingface.co/sentence-transformers/clip-ViT-B-32-multilingual-v1

[NO.2021-0-01343, Artificial Intelligence Graduate School Program (Seoul National University) & NO.2021-0-02068, Artificial Intelligence Innovation Hub (Artificial Intelligence Institute, Seoul National University)].

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

## A  Pre-training Details

Like Fredrik Carlsson (2022)'s method, our multilingual CLIP is trained by pre-training through teacher learning using MSE loss as shown on the left of the Figure 2. The datasets used for pretraining are GCC (Wang et al., 2019), VizWiz, and MSCOCO with total 2.2M sentences. Each English sentence is translated into German, Spanish, French, and Japanese using MBART-50. The pretrained CLIP model used as the teacher model is the RN50X4 model, and the Distill-Multilingual BERT (Sanh et al., 2019) is used as the student text encoder. The model is trained in a total of 5 epochs, and it takes about 20 hours per epoch with our computing power .

## B  Experimental Details

### B.1  Reproductabilty checklists

**Dataset and Source code** We provide our pretraining, fine-tuning, and evaluation source code along with configuration code for perturbations as supplementary materials. We will publicly release our dataset M-FineCap3k, and the full codes with weight parameters.

**Computing Resources** AMD Ryzen Threadripper 2950X (3.50 GHz) with GeForce GTX 2080 Ti is used for the experiments. All codes are implemented on Python 3.6.15 and PyTorch 1.7.1. The fine-tuning of each model trains 5 epochs, and takes about 6 hours per epoch.

**Number of Parameters** The number of parameter of our multilingual CLIP is about 66M as like as Distill-Multilingual BERT.

**Train-Valid-Test split** MSCOCO used for finetuning consists of 414k training set and 25k validation set. We split the training set by 9:1 and used it for fine-tuning and validation. We also randomly extracted 3k samples from the existing validation set and used it as a test set.

### B.2  Hyper-parameters

**Hyper-parameters for fine-tuning** In order to find the best-performing model, we conducted an experiment on 16 hyper-parameter combinations ($\lambda_1 : 0 \sim 0.5, \lambda_2 : 0 \sim 0.1, \lambda_3 : 0 \sim 0.1$). The hyper-parameter was manually tuned based on the effective detection of lexical noise while maintaining high human correlation, and finally, the best-performing $\lambda$ values of the objective function for fine-tuning are as follows: $\lambda_1 = 0.1, \lambda_2 = 0.05, \lambda_3 = 0.05$. In the main text, we reported single-run scores after finding the best performing parameters.

**Hyper-parameters for optimizer** We use AdamW (Loshchilov and Hutter, 2017) optimizer with $\beta_1 = 0.9, \beta_2 = 0.999, \epsilon = 1e-8$. The initial learning rate is $5e-5$.

## C  M-CapEval1k examples

The examples of the M-CapEval1k can be seen in Figure 8 and Figure 9. Native speakers of each language translated into each language while maintaining the score based on the image, original caption, and score assigned to the pair. The word transcreate is used because it was translated while maintaining the score, not a simple translation. The API for collecting the M-CapEval1k dataset is shown in Figure 10, and instructions for collecting the dataset can be show in Figure 11.

## D  M-FineCap3k statistics

| Language | EN | DE | ES | FR | JA |
|---|---|---|---|---|---|
| **Datasize** | 1,000 | 3,000 | 3000 | 3,000 | 3,000 |
| **Sentence length** | 23.42 | 19.42 | 23.21 | 22.09 | 48.39 |
| **# of critical objects** | 2.87 | 3.56 | 4.17 | 3.11 | 4.42 |
| **# of backgrounds** | 1.25 | 1.27 | 1.39 | 1.25 | 1.56 |
| **# of relationships** | 1.57 | 2.05 | 1.78 | 1.81 | 2.46 |

Table 5: M-FineCap3k statistics

Table 5 provides detailed statistics for M-FineCap3k, including the dataset size for each language, the average sentence length, and the average numbers of critical objects, backgrounds, and relationships. M-FineCap3k consists of lengthy sentences of approximately 20 word tokens on average. In the case of Japanese, since there is no spacing in a sentence, sentence length is calculated using a tokenizer based on word extractor, and the sentence length is almost the same as the character level. In addition, there are three to four critical objects in all languages, so each sentence describes the visual content of an image in great detail.

## E M-FineCap3k eval set examples

The examples of the M-FineCap3k eval set for languages other than English can be seen in Figure 7.

**French**
**Caption:** Une femme attend en ligne à côté d'une femme avec un sac marron dans une pièce bondée.
**Critical Objects** : femme
**Relationships** : attend
**Backgrounds** : pièce bondée

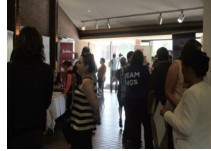

**German**
**Caption:** Vor einem grauen Hintergrund befindet sich eine lächelnde Frau, sie trägt eine goldene Halskette und hat grüngraue Augen.
**Critical Objects** : eine lächelnde Frau, goldene Halskette, grüngraue Augen
**Relationships** : befindet sich, trägt
**Backgrounds** : grauen Hintergrund

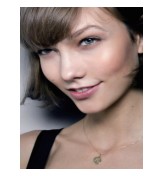

**Spanish**
**Captions:** grupo de personas sonriendo y haciendo un gesto con las manos, al fondo un panel blanco con diferentes logos
**Critical Objects** : grupo de personas, gesto, manos, logos
**Relationships** : sonriendo, haciendo
**Backgrounds** : panel blanco

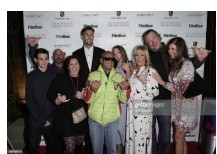

**Japanese**
**Captions:** 白い柱と茶色いフェンスに囲まれたテラスに白いクッションの椅子2脚と鉢植えが並んでいて、クッションが乗った白いソファと銀色の屋外用ストーブがあり、後ろに木と砂浜と海と早朝か夕方の空が見える。
**Critical Objects** : 白い柱, 茶色いフェンス, テラス, 白いクッションの椅子2脚, 鉢植え, クッション, 白いソファ, 銀色の屋外用ストーブ
**Relationships** : 囲まれた, 乗った
**Backgrounds** : 木, 砂浜, 海, 早朝か夕方の空

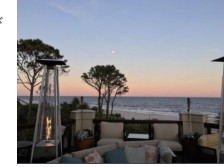

Figure 7: M-FineCap3k eval set examples for each languages.

## F Perturbed caption examples

The examples of the perturbed captions for languages other than English can be seen in Figure 12-Figure 15. The critical objects shuffled for in-sentence substitution perturbation are displayed using each color.

## G Implementation Details

In Alg. 1, we show the Python implementation of each perturbation criterion: *"Repetition"*, *"Removal"*, *"Masking"*, *"Jumble"*, and *"Substitution"*.

## H All results tables

**MSCOCO** The results for all perturbation of all languages for MSCOCO 3k eval set can be found in Table 7.

**Flickr8k** The results for all perturbation of all languages for Flickr8k eval set can be found in Table 8.

**VizWiz** The results for all perturbation of all languages for Vizwiz eval set can be found in Table 9.

**Multi30k** The results for all perturbation of all languages for Multi30k eval set can be found in Table 10.

**M-FineCap3k** The results for all perturbation of all languages for M-FineCap3k eval set can be found in Table 11.

## I Correlations with human judgement

| Metric | DE | | FR | | ES | |
|---|---|---|---|---|---|---|
| | Composite | Flickr8k | Composite | Flickr8k | Composite | Flickr8k |
| BLEU-1 | 0.301 | 0.313 | 0.302 | 0.314 | 0.303 | 0.313 |
| BLEU-4 | 0.299 | 0.301 | 0.303 | 0.302 | 0.304 | 0.303 |
| ROUGE-L | 0.315 | 0.309 | 0.317 | 0.311 | 0.320 | 0.312 |
| CIDEr | 0.377 | 0.443 | 0.376 | 0.435 | 0.377 | 0.432 |
| MCS | **0.498** | **0.528** | **0.497** | **0.530** | **0.496** | **0.529** |
| PR-MCS | **0.494** | **0.524** | **0.496** | **0.527** | **0.496** | **0.528** |

Table 6: Kendall Tau ($\tau_c$) correlations with human judgment for existing datasets other than English.

The results for diverse languages on the Composite (Aditya et al., 2015) and Flickr8k_Expert (Hodosh et al., 2013) datasets are shown in the Table 6. Given that these benchmark datasets are initially in English, we employed the MBART-50 machine translation model (Tang et al., 2020) to translate them into various languages. When contrasted with reference-based image captioning metrics, PR-MCS showcases a significantly strong correlation across all languages for both datasets, and its performance closely resembles that of MCS in a comparative analysis.

**Algorithm 1** Python implementation of perturbation

```python
def rep_rem_mask(caption_list): # Repetition, Removal, and Masking
    caption_rp = []
    caption_rm = []
    caption_rm_mask = []

    for i in range(len(caption_list)):
        words = caption_list[i].split()
        substitued_rp = []
        substitued_rm = []
        substitued_rm_mask = []
        substitued_rmrp = []
        for j in range(len(words)):
            substitued_rp.append(words[j])
            substitued_rm_mask.append(words[j])
            if random.random() > threshold:
                substitued_rp.append(words[j])
                substitued_rm.append(words[j])
                substitued_rm_mask[-1] = '[MASK]'
            elif random.random() > threshold:
                substitued_rmrp.append(words[j])
        caption_rp.append(" ".join(substitued_rp))
        caption_rm.append(" ".join(substitued_rm))
        caption_rm_mask.append(" ".join(substitued_rm_mask))

    return caption_rp, caption_rm, caption_rm_mask

def jumble(caption_list): # Jumble
    caption_jumble = []
    for i in range(len(caption_list)):
        words = caption_list[i].split()
        random.shuffle(words)
        caption_jumble.append(" ".join(words))
    return caption_jumble

def sub_in_sent(caption_list, critical_obj_list): # In-sentence substitution
    caption_sub_in = []
    for i in range(len(caption_list)):
        current_caption = caption_list[i]
        current_critical_obj_list = critical_obj_list[i]
        shuffled = current_critical_obj_list.copy()
        words = caption_list[i].split()

        if len(current_critical_obj_list) < 2:
            caption_sub_in.append(" ".join(words))
        else:
            while current_critical_obj_list == shuffled:
                random.shuffle(shuffled)

            target = current_caption
            for j in range(len(shuffled)):
                target = shuffled[j].join(target.rsplit(current_critical_obj_list[j],1))
            caption_sub_in.append(target)

    return caption_sub_in
```

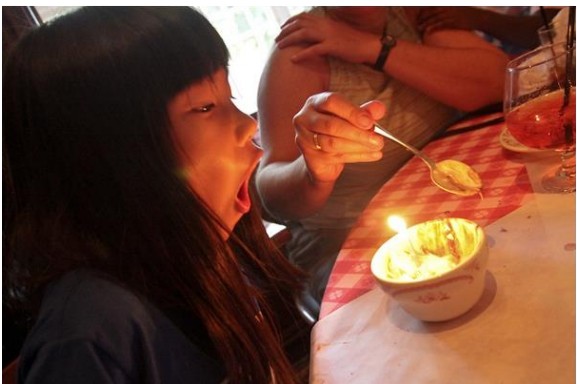

## Original Captions

a woman sitting at a table with a cake

## Score (0-1 range)

0.6

## Transcreated Captions

(FR) une femme assise à une table avec un gâteau

(DE) eine Frau sitzt an einem Tisch mit einem Kuchen

(ES) una mujer sentada en la mesa con un pastel

(JA) ケーキの載ったテーブルの横に座っている1人の女性

Figure 8: M-CapEval1k example (1).

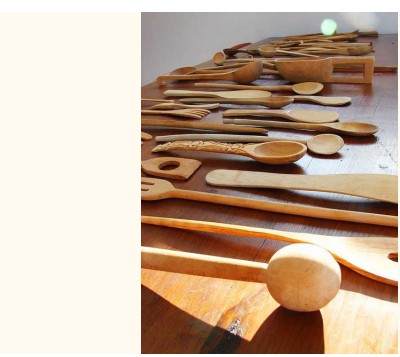

## Original Captions

a bunch of knives on a cutting board with a knife

## Score (0-1 range)

0.2

## Transcreated Captions

(FR) un tas de couteaux sur une planche à découper avec un couteau

(DE) einige Messer auf einem Schneidebrett mit einem Messer

(ES) un conjunto de cuchillos sobre una tabla de cortar con un cuchillo

(JA) まな板の上のたくさんのナイフと1本のナイフ

Figure 9: M-CapEval1k example (2).

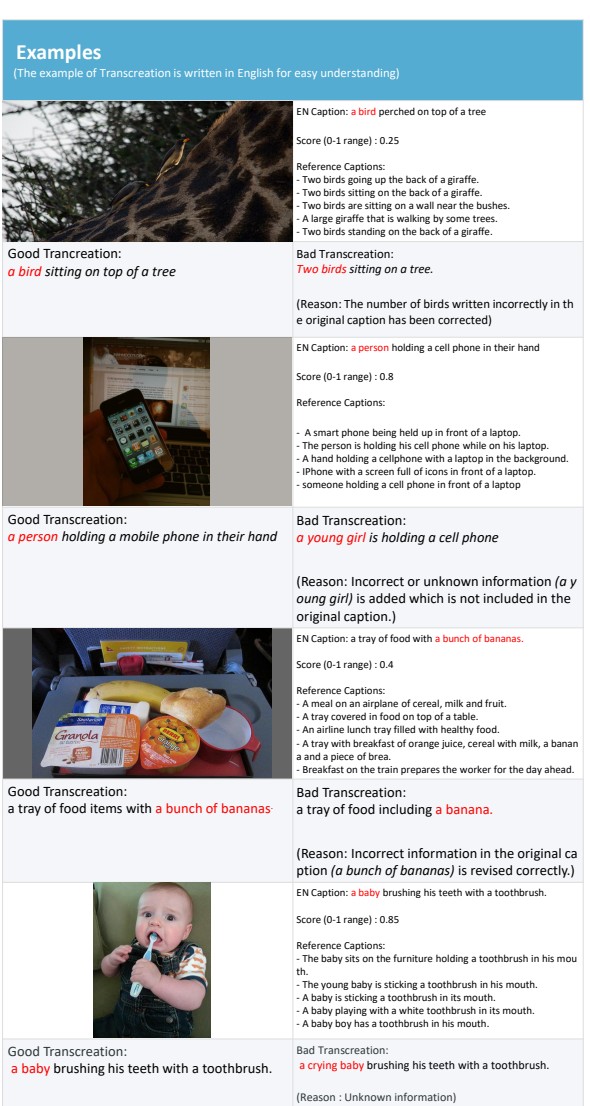

Figure 10: M-CapEval1k collection API.

Figure 11: M-CapEval1k collecting instructions.

## French

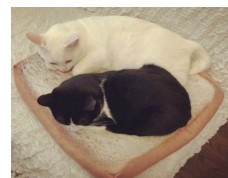

**Critical Objects :** chat noir, chat blanc, matelas pour animaux de compagnie blanc et rose

**Original:** Un chat noir et un chat blanc allongés ensemble sur un matelas pour animaux de compagnie blanc et rose posé sur un tapis blanc.

**Jumble:** pour Un compagnie un blanc un sur chat blanc sur rose et matelas de allongés un blanc. posé animaux et noir tapis ensemble chat

**Removal:** Un et un chat blanc allongés un pour animaux compagnie rose posé blanc.

**Repetition:** Un Un chat noir et et un un chat chat blanc blanc allongés allongés ensemble sur un un matelas pour pour animaux animaux de compagnie compagnie blanc et rose rose posé posé sur un tapis blanc.

**Removal_mask:** [MASK] chat noir [MASK] [MASK] [MASK] [MASK] ensemble sur [MASK] matelas [MASK] [MASK] de [MASK] blanc et [MASK] [MASK] sur un tapis [MASK]

**Sub_in_sent:** Un chat blanc et un matelas pour animaux de compagnie blanc et rose allongés ensemble sur un chat noir posé sur un tapis blanc.

Figure 12: Eval set perturbed captions example (FR).

## Japanese

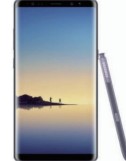

**Critical Objects :** 黒いフレームのスマホ, 空と海のような青色と黄色, 黒い陸, 人影, 紫色のタッチペン, スマホ

**Original:** 無地の白色を背景として、黒いフレームのスマホがあり、画面には空と海のような青色と黄色に黒い陸と人影が浮かんでいて、紫色のタッチペンがスマホに立てかけられている。

**Jumble:** いる色青タッチペンと黄色かけし画面の空は浮かんられフレームスマホをてにのにと黒い色と黒い海ありが。スマホとがようの、にいが背景、、て立て紫陸てた人影の白色無地で

**Removal:** 無地の白色をして、あり、海青と黒いが浮かんでて色て

**Repetition:** 無地無地のの白色白色をを背景としして、、黒いフレームのスマホがありあり、、画面には空と海海のような青色ととと黄色に黒い黒い陸と人影がが浮かん浮かんでいてて、紫色色のタッチペンがスマホに立てかけられてている。

**Removal_mask:** [MASK][MASK][MASK][MASK]背景と[MASK][MASK][MASK]黒いフレームのスマホが[MASK][MASK]画面には空と[MASK]のような[MASK]色[MASK]黄色に[MASK]陸と人影[MASK][MASK][MASK]い[MASK]、紫[MASK]のタッチペンがスマホに立てかけられ[MASK]いる。

**Sub_in_sent:** 無地の白色を背景として、空と海のような青色と黄色があり、画面には黒い陸に黒いフレームのスマホと紫色のタッチペンが浮かんでいて、スマホが人影に立てかけられている。

Figure 14: Eval set perturbed captions example (JA).

## German

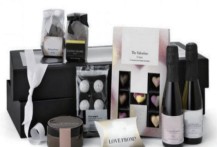

**Critical Objects :** Packungen, Flaschen, Pralinen

**Original:** Vor einem weißen Hintergrund stehen mehrere Packungen und Flaschen, in den Packungen befinden sich Pralinen

**Jumble:** einem und Hintergrund stehen weißen Packungen Flaschen, sich Packungen mehrere vor in Pralinen den befinden

**Removal:** vor weißen Packungen Pralinen

**Repetition:** vor vor einem weißen weißen Hintergrund stehen mehrere Packungen und Flaschen, in den Packungen Packungen befinden sich Pralinen Pralinen

**Removal_mask:** [MASK] einem [MASK] Hintergrund stehen mehrere Packungen und Flaschen, in den [MASK] [MASK] [MASK] befinden sich [MASK]

**Sub_in_sent:** vor einem weißen Hintergrund stehen mehrere Packungen und Pralinen, in den Packungen befinden sich Flaschen

Figure 13: Eval set perturbed captions example (DE).

## Spanish

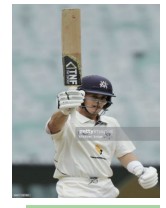

**Critical Objects :** Hombre, uniforme deportivo blanco, casco con protector facial, guantes

**Original:** Hombre vistiendo un uniforme deportivo blanco con casco con protector facial y con guantes blancos, sosteniendo en la mano derecha un palo de cricket con fondo claro difuso

**Jumble:** blancos, con sosteniendo en uniforme guantes difuso derecha claro con protector casco Hombre blanco con palo un vistiendo de cricket un mano y la facial deportivo con fondo

**Removal:** vistiendo uniforme con casco facial blancos, en con fondo claro difuso

**Repetition:** Hombre vistiendo vistiendo un uniforme uniforme deportivo blanco con con casco casco con protector facial facial y con guantes blancos, blancos, sosteniendo en en la mano derecha un palo de cricket con con fondo fondo claro claro difuso difuso

**Removal_mask:** Hombre [MASK] un [MASK] deportivo blanco [MASK] [MASK] con protector [MASK] y con guantes [MASK] sosteniendo [MASK] la mano derecha un palo de cricket [MASK] [MASK] [MASK] [MASK]

**Sub_in_sent:** palo de cricket vistiendo un casco con protector facial con y guantes con uniforme deportivo blanco, sosteniendo en la mano derecha un Hombre blancos con fondo claro difuso

Figure 15: Eval set perturbed captions example (ES).

| Language | Metric | Original | Repetition | Removal | Masking | Jumble | Substitution | Average |
|---|---|---|---|---|---|---|---|---|
| **English** | †**MCS** | 0.7944 | 0.7871 (-0.92%) | 0.7241 (-8.84%) | 0.7137 (-10.15%) | 0.7573 (-4.67%) | 0.7817 (-1.59%) | 0.7442 (-5.23%) |
| | *****MCS** | 1.0939 | 1.0963 (0.21%) | 1.0177 (-6.96%) | 0.987 (-9.77%) | 1.0966 (-0.24%) | 1.0928 (-0.10%) | 1.05808 (-3.27%) |
| | **PR-MCS** | 1.4177 | 0.8800 (-37.93%) | 0.3103 (-78.11%) | 0.0112 (-99.21%) | 0.0433 (-96.95%) | 0.2534 (-82.13%) | **0.29964** **(-78.86%)** |
| **German** | †**MCS** | 1.0743 | 1.0764 (0.19%) | 0.9989 (-7.01%) | 0.9501 (-11.56%) | 1.0792 (0.45%) | 1.0735 (-0.07%) | 1.03562 (-3.60%) |
| | *****MCS** | 1.0839 | 1.0851 (-0.11%) | 1.0119 (-6.64%) | 1.0044 (-7.33%) | 1.0887 (0.44%) | 1.0855 (0.15%) | 1.05512 (-2.66%) |
| | **PR-MCS** | 1.3153 | 0.9082 (-30.95%) | 0.686 (-47.84%) | 0.0175 (-98.67%) | 0.166 (-87.38%) | 0.8004 (-39.15%) | **0.51562** **(-60.80%)** |
| **French** | †**MCS** | 1.0719 | 1.0753 (0.32%) | 0.9983 (-6.87%) | 0.9453 (-11.81%) | 1.0768 (0.46%) | 1.0692 (-0.25%) | 1.03298 (-3.63%) |
| | *****MCS** | 1.0837 | 1.0845 (0.07%) | 1.0152 (-6.32%) | 0.9918 (-8.48%) | 1.0878 (0.37%) | 1.0857 (0.18%) | 1.053 (-2.83%) |
| | **PR-MCS** | 1.4127 | 1.0021 (-29.21%) | 0.9086 (-35.68%) | 0.0091 (-99.36%) | 0.2873 (-76.66%) | 1.1167 (-20.95%) | **0.66434** **(-52.97%)** |
| **Spanish** | †**MCS** | 1.0564 | 1.0591 (0.26%) | 0.9865 (-6.61%) | 0.9331 (-11.67%) | 1.0624 (0.57%) | 1.0528 (-0.34%) | 1.01878 (-3.56%) |
| | *****MCS** | 1.0799 | 1.0807 (0.07%) | 1.012 (-6.29%) | 0.9909 (-8.24%) | 1.0846 (0.44%) | 1.0812 (0.12%) | 1.04988 (-2.77%) |
| | **PR-MCS** | 1.4086 | 1.0316 (-26.76%) | 0.8841 (-37.24%) | 0.0101 (-99.28%) | 0.2882 (-79.54%) | 1.193 (-15.31%) | **0.6814** **(-51.63%)** |
| **Japanese** | †**MCS** | 1.0727 | 1.0769 (0.39%) | 1.0001 (-6.68%) | 0.9247 (-13.80%) | 1.0819 (0.86%) | 1.0726 (0.01%) | 1.03122 (-3.85%) |
| | *****MCS** | 1.0286 | 1.0348 (0.60%) | 0.9669 (-6.00%) | 0.9105 (-11.48%) | 1.0294 (0.08%) | 1.0271 (-0.15%) | 0.9813 (-3.39%) |
| | **PR-MCS** | 1.3948 | 1.3114 (-5.97%) | 1.0539 (-24.44%) | 0.0116 (-99.16%) | 0.9114 (-34.66%) | 1.3128 (-5.88%) | **0.92022** **(-34.02%)** |

† : Fredrik Carlsson (2022) based

*: Our Multilingual CLIP based

Table 7: MSCOCO 3k results table.

| Language | Metric | Original | Repetition | Removal | Masking | Jumble | Substitution | Average |
|---|---|---|---|---|---|---|---|---|
| **English** | †**MCS** | 1.0659 | 1.0562 (-0.91%) | 0.9882 (-7.29%) | 0.8991 (-15.65%) | 1.0524 (-1.27%) | 1.0611 (-0.45%) | 1.0114 (-5.11%) |
| | *****MCS** | 1.0915 | 1.0919 (0.04%) | 1.0102 (-7.45%) | 0.9839 (-9.86%) | 1.0952 (0.34%) | 1.0903 (-0.11%) | 1.0543 (-3.41%) |
| | **PR-MCS** | 1.1484 | 0.7679 (-33.13%) | 0.2721 (-76.31%) | 0.0065 (-99.43%) | 0.0163 (-98.58%) | 0.138 (-87.98%) | 0.24016 **(-79.09%)** |
| **German** | †**MCS** | 1.0688 | 1.067 (-0.17%) | 0.9852 (-7.82%) | 0.9059 (-15.24%) | 1.0738 (0.47%) | 1.0664 (-0.22%) | 1.01966 (-4.60%) |
| | *****MCS** | 1.0779 | 1.0763 (-0.15%) | 0.9958 (-7.62%) | 0.9921 (-7.96%) | 1.0839 (0.56%) | 1.0796 (0.16%) | 1.04554 (-3.00%) |
| | **PR-MCS** | 1.1621 | 0.7511 (-35.37%) | 0.3756 (-67.68%) | 0.0203 (-98.25%) | 0.0346 (-97.02%) | 0.1556 (-86.61%) | 0.26744 **(-76.99%)** |
| **French** | †**MCS** | 1.0653 | 1.0636 (-0.16%) | 0.9815 (-7.87%) | 0.8863 (-16.80%) | 1.0625 (-0.26%) | 1.0594 (-0.55%) | 1.01066 (-5.13%) |
| | *****MCS** | 1.0758 | 1.0745 (-0.12%) | 0.9986 (-7.18%) | 0.9855 (-8.39%) | 1.0824 (0.61%) | 1.0778 (0.19%) | 1.04376 (-2.98%) |
| | **PR-MCS** | 1.6681 | 1.0369 (-37.84%) | 0.6367 (-61.83%) | 0.018 (-98.92%) | 0.0545 (-96.73%) | 0.3915 (-76.53%) | 0.42752 **(-74.37%)** |
| **Spanish** | †**MCS** | 1.0527 | 1.0517 (-0.09%) | 0.9707 (-7.79%) | 0.8773 (-16.66%) | 1.0445 (-0.78%) | 1.0451 (-0.72%) | 0.99786 (-5.21%) |
| | *****MCS** | 1.0747 | 1.0732 (-0.14%) | 0.999 (-7.04%) | 0.9829 (-8.54%) | 1.0802 (0.51%) | 1.0756 (0.08%) | 1.04218 (-3.03%) |
| | **PR-MCS** | 1.6534 | 1.0358 (-37.35%) | 0.6189 (-62.57%) | 0.0111 (-99.33%) | 0.0587 (-96.45%) | 0.618 (-62.62%) | 0.4685 **(-71.66%)** |
| **Japanese** | †**MCS** | 1.0703 | 1.0722 (0.18%) | 0.9875 (-7.74%) | 0.8573 (-19.90%) | 1.0796 (0.87%) | 1.067 (-0.31%) | 1.01272 (-5.38%) |
| | *****MCS** | 1.0293 | 1.0275 (-0.17%) | 0.9655 (-6.20%) | 0.8932 (-13.22%) | 1.0278 (-0.15%) | 1.0255 (-0.37%) | 0.9879 (-4.02%) |
| | **PR-MCS** | 1.6279 | 0.7931 (-51.28%) | 0.6235 (-61.70%) | 0.0106 (-99.35%) | 0.0894 (-94.51%) | 0.715 (-56.08%) | 0.44632 **(-72.58%)** |

† : Fredrik Carlsson (2022) based

*: Our Multilingual CLIP based

Table 8: Flickr8k results table.

| Language | Metric | Original | Repetition | Removal | Masking | Jumble | Substitution | Average |
|----------|--------|----------|------------|---------|---------|--------|--------------|---------|
| **English** | †**MCS** | 0.7217 | 0.7134 (-1.15%) | 0.6763 (-6.29%) | 0.625 (-13.4%) | 0.7007 (-2.91%) | 0.7114 (-1.43%) | 0.68536 (-5.04%) |
|  | *****MCS** | 1.0626 | 1.0606 (-0.19%) | 1.0063 (-5.3%) | 0.9756 (-8.19%) | 1.0651 (0.24%) | 1.0614 (-0.11%) | 1.0338 (-2.71%) |
|  | **PR-MCS** | 0.9769 | 0.6281 (-35.7%) | 0.2754 (-71.81%) | 0.0069 (-99.29%) | 0.0892 (-90.87%) | 0.4726 (-51.62%) | **0.29444** **(-69.86%)** |
| **German** | †**MCS** | 1.0284 | 1.0213 (-0.69%) | 0.9757 (-5.12%) | 0.9176 (-10.77%) | 1.0352 (0.66%) | 1.0254 (-0.29%) | 0.99504 (-3.24%) |
|  | *****MCS** | 1.0491 | 1.0433 (-0.55%) | 0.9952 (-5.14%) | 0.9848 (-6.13%) | 1.0515 (0.23%) | 0.9754 (-7.03%) | 1.01004 (-3.72%) |
|  | **PR-MCS** | 0.9895 | 0.6857 (-30.7%) | 0.4573 (-53.78%) | 0.0154 (-98.44%) | 0.1558 (-84.25%) | 0.7607 (-23.12%) | **0.41498** **(-58.06%)** |
| **French** | †**MCS** | 1.0266 | 1.0206 (-0.58%) | 0.9788 (-4.66%) | 0.9034 (-12%) | 1.0369 (1%) | 1.0228 (-0.37%) | 0.9925 (-3.32%) |
|  | *****MCS** | 1.0509 | 1.0456 (-0.5%) | 1.0001 (-4.83%) | 0.9764 (-7.09%) | 1.0534 (0.24%) | 1.0499 (-0.1%) | 1.02508 (-2.46%) |
|  | **PR-MCS** | 1.442 | 0.9655 (-33.04%) | 0.7088 (-50.85%) | 0.0097 (-99.33%) | 0.2314 (-83.95%) | 1.166 (-19.14%) | **0.61628** **(-57.26%)** |
| **Spanish** | †**MCS** | 1.0226 | 1.0189 (-0.36%) | 0.9794 (-4.22%) | 0.8924 (-12.73%) | 1.0377 (1.48%) | 1.0207 (-0.19%) | 0.98982 (-3.21%) |
|  | *****MCS** | 1.0505 | 1.0463 (-0.4%) | 1.002 (-4.62%) | 0.973 (-7.38%) | 1.0537 (0.3%) | 0.9982 (-4.98%) | 1.01464 (-3.41%) |
|  | **PR-MCS** | 1.4526 | 0.9893 (-31.89%) | 0.6767 (-53.41%) | 0.003 (-99.79%) | 0.209 (-85.61%) | 1.2505 (-13.91%) | **0.6257** **(-56.93%)** |
| **Japanese** | †**MCS** | 1.0297 | 1.0279 (-0.17%) | 0.9855 (-4.29%) | 0.8733 (-15.19%) | 1.05 (1.97%) | 1.0259 (-0.37%) | 0.99252 (-3.61%) |
|  | *****MCS** | 1.0414 | 1.0242 (-1.65%) | 0.9963 (-4.33%) | 0.9275 (-10.94%) | 1.0406 (-0.08%) | 1.0403 (-0.11%) | 1.00578 (-3.42%) |
|  | **PR-MCS** | 1.4113 | 0.8612 (-38.98%) | 0.7769 (-44.95%) | 0.0035 (-99.75%) | 0.3241 (-77.04%) | 1.1465 (-18.76%) | **0.62244** **(-55.9%)** |

† : Fredrik Carlsson (2022) based

*: Our Multilingual CLIP based

Table 9: VizWiz results table.

| Language | Metric | Original | Repetition | Removal | Masking | Jumble | Substitution | Average |
|---|---|---|---|---|---|---|---|---|
| **English** | †**MCS** | 0.8234 | 0.8201 (-0.40%) | 0.7613 (-7.54%) | 0.7896 (-4.10%) | 0.8221 (-1.58%) | 0.7968 (-3.23%) | 0.7978 (-3.09%) |
| | ***MCS** | 1.0611 | 1.0568 (-0.41%) | 0.9965 (-6.09%) | 0.9801 (-7.63%) | 1.058 (-0.29%) | 0.9827 (-7.39%) | 1.01482 (-4.36%) |
| | **PR-MCS** | 1.1008 | 0.7963 (-27.66%) | 0.2309 (-79.01%) | 0.0095 (-99.13%) | 0.032 (-97.09%) | 0.4516 (-58.97%) | **0.30406** **(-72.37%)** |
| **German** | †**MCS** | 1.0325 | 1.0298 (-0.26%) | 0.9863 (-4.47%) | 0.9691 (-6.14%) | 1.0308 (0.16%) | 0.976 (-5.47%) | 0.9984 (-3.30%) |
| | ***MCS** | 1.0625 | 1.0581 (-0.41%) | 0.9809 (-7.68%) | 0.9737 (-8.36%) | 1.0606 (0.18%) | 1.0586 (-0.36%) | 1.02638 (-3.40%) |
| | **PR-MCS** | 1.096 | 0.7419 (-32.3%) | 0.3583 (-67.31%) | 0.0123 (-98.88%) | 0.0544 (-95.04%) | 0.6396 (-41.64%) | **0.3613** **(-67.03%)** |
| **French** | †**MCS** | 1.0353 | 1.031 (-0.42%) | 1.0098 (-2.46%) | 0.9782 (-5.51%) | 1.0329 (-0.23%) | 0.9853 (-4.82%) | 1.00744 (-2.69%) |
| | ***MCS** | 1.0624 | 1.0581 (-0.40%) | 0.9848 (-7.30%) | 0.9651 (-9.15%) | 1.058 (-0.41%) | 1.0578 (-0.43%) | 1.02476 (-3.54%) |
| | **PR-MCS** | 1.5767 | 0.9374 (-40.54%) | 0.590 (-62.58%) | 0.0192 (-98.78%) | 0.052 (-96.70%) | 0.9585 (-39.21%) | **0.51142** **(-67.56%)** |

† : Fredrik Carlsson (2022) based

\*: Our Multilingual CLIP based

Table 10: Multi30K results table.

| Language | Metric | Original | Repetition | Removal | Masking | Jumble | Substitution | Average |
|---|---|---|---|---|---|---|---|---|
| **English** | †MCS | 0.7593 | 0.7524 (-0.91%) | 0.6577 (-13.38%) | 0.6265 (-17.49%) | 0.7275 (-4.19%) | 0.7515 (-1.03%) | 0.70312 (-7.40%) |
| | *MCS | 1.0657 | 1.0439 (-2.05%) | 0.9699 (-8.99%) | 1.0484 (-1.62%) | 1.0628 (-0.27%) | 1.061 (-0.44%) | 1.0372 (-2.67%) |
| | PR-MCS | 1.0429 | 0.5984 (-42.62%) | 0.2382 (-77.16%) | 0.4844 (-53.55%) | 0.7262 (-30.37%) | 0.7421 (-28.84%) | **0.55786** **(-46.51%)** |
| | PR-MCS (Few-shot) | 0.7125 | 0.3183 (-55.33%) | 0.0653 (-90.84%) | 0.0012 (-99.83%) | 0.0027 (-99.62%) | 0.4045 (-43.23%) | **0.1584** **(-77.77%)** |
| **German** | †MCS | 1.0708 | 1.0576 (-1.23%) | 0.9516 (-11.13%) | 1.0011 (-6.51%) | 1.0546 (-1.51%) | 1.0643 (-0.61%) | 1.02584 (-4.20%) |
| | *MCS | 1.0803 | 1.063 (-1.60%) | 0.9638 (-10.78%) | 1.0484 (-2.95%) | 1.0815 (0.11%) | 1.0781 (-0.20%) | 1.04696 (-3.09%) |
| | PR-MCS | 0.7261 | 0.5437 (-25.12%) | 0.2521 (-65.28%) | 0.4996 (-31.19%) | 0.5151 (-29.06%) | 0.495 (-31.83%) | **0.4611** **(-36.5%)** |
| | PR-MCS (Few-shot) | 0.589 | 0.396 (-32.77%) | 0.0617 (-89.52%) | 0.0011 (-99.81%) | 0.1059 (-82.02%) | 0.3062 (-48.01%) | **0.17418** **(-70.43%)** |
| **French** | †MCS | 1.053 | 1.045 (-0.76%) | 0.9462 (-10.14%) | 0.991 (-5.89%) | 1.048 (-0.47%) | 1.0463 (-0.64%) | 1.0153 (-3.58%) |
| | *MCS | 1.0575 | 1.0472 (-0.97%) | 0.957 (-9.50%) | 1.0322 (-2.39%) | 1.0574 (-0.01%) | 1.0536 (-0.37%) | 1.02948 (-2.65%) |
| | PR-MCS | 1.3634 | 0.96 (-29.59%) | 0.4739 (-65.24%) | 1.0976 (-19.50%) | 0.7335 (-46.2%) | 0.9791 (-28.19%) | **0.84882** **(-37.74%)** |
| | PR-MCS (Few-shot) | 1.4862 | 1.025 (-31.03%) | 0.1559 (-89.51%) | 0.0082 (-99.45%) | 0.1034 (-93.04%) | 0.9032 (-39.23%) | **0.43914** **(-70.45%)** |
| **Spanish** | †MCS | 1.0599 | 1.048 (-1.12%) | 0.9548 (-9.92%) | 0.9996 (-5.69%) | 1.0464 (-1.27%) | 1.0536 (-0.59%) | 1.02048 (-3.72%) |
| | *MCS | 1.0577 | 1.0385 (-1.82%) | 0.9629 (-8.96%) | 1.0393 (-1.74%) | 1.0542 (-0.33%) | 1.0537 (-0.38%) | 1.02972 (-2.65%) |
| | PR-MCS | 1.3821 | 1.206 (-12.74%) | 0.5811 (-57.96%) | 1.1648 (-15.72%) | 0.9128 (-33.96%) | 0.8481 (-38.64%) | **0.94256** **(-31.80%)** |
| | PR-MCS (Few-shot) | 1.4292 | 1.187 (-16.95%) | 0.1798 (-87.42%) | 0.0019 (-99.87%) | 0.1405 (-90.17%) | 0.473 (-66.90%) | **0.39644** **(-72.26%)** |
| **Japanese** | †MCS | 1.06 | 1.0546 (-0.51%) | 0.9768 (-7.85%) | 0.9976 (-5.89%) | 1.0475 (-1.18%) | 1.0559 (-0.39%) | 1.02648 (-3.16%) |
| | *MCS | 0.963 | 0.9692 (0.64%) | 0.982 (1.97%) | 1.0323 (7.20%) | 0.9813 (1.90%) | 0.963 (0.00%) | 0.98556 (2.34%) |
| | PR-MCS | 1.1551 | 0.2719 (-76.46%) | 0.3357 (-70.94%) | 0.7354 (-36.33%) | 0.6373 (-44.83%) | 1.0862 (-5.96%) | **0.6133** **(-46.91%)** |
| | PR-MCS (Few-shot) | 1.4136 | 0.1385 (-90.2%) | 0.1285 (-90.91%) | 0.0012 (-99.92%) | 0.0048 (-99.66%) | 1.2969 (-8.26%) | **0.31398** **(-77.79%)** |

† : Fredrik Carlsson (2022) based

*: Our Multilingual CLIP based

Table 11: M-FineCapEval results table.