# OpenReview forum: "PR-MCS: Perturbation Robust Metric for MultiLingual Image Captioning"
_EMNLP/2023/Conference — EMNLP 2023 Findings_

### Official Review · Reviewer_ccYd · 2023-08-02

**Soundness:** 3

**Excitement:**

3: Ambivalent: It has merits (e.g., it reports state-of-the-art results, the idea is nice), but there are key weaknesses (e.g., it describes incremental work), and it can significantly benefit from another round of revision. However, I won't object to accepting it if my co-reviewers champion it.

**Paper Topic And Main Contributions:**

- They finetune the text encoder of CLIP to distinguish the perturbed text from the original caption.
- They introduce a new evaluation dataset, M-FineCap3k, containing 5 translated languages from English captions.
- The proposed PR-MCS can distinguish lexically perturbed text in different datasets.

**Questions For The Authors:**

- From Table 1, the scores of the MCS(dagger) and MCS(star) are almost same. What is the advantage of doing the pre-training? The MCS(dagger) already provides pre-trained models with different languages.
- For the loss of PR-MCS, the idea is to make the distance between the image embedding and the original caption embedding close and the distance between the perturbed caption embedding and other embeddings further. I wonder if the Triplet loss can be used instead of pair-wise loss. Have you tried or thought about the triplet loss when you design the model?
- In Table 2, the few shot results are based on the different test sets on the regular M-FineCap3k dataset. It can mislead the performance results by reporting under the same dataset. Also, how many few shots are used for the evaluation? It would be more informative if the statistics of the sequence length on the datasets is provided for L510-L516.
- In Table 3, PR-MCS results underperform in general except for the Pearson correlation in English. Also, can you provide the significant test results for the correlations? It does not provide convincing evidence to claim that PR-MCS is a metric with a strong correlation with human judgment.

**Reasons To Accept:**

- The paper shows the proper motivation for the proposed datasets and reasonable method and experimental setup.
- Though contrastive loss is not a novel method for multi-lingual settings, pair-wise contrastive loss for different languages can have benefits.
- The experiments are performed on different datasets.

**Reasons To Reject:**

Please see the questions below.

**Reproducibility:**

4: Could mostly reproduce the results, but there may be some variation because of sample variance or minor variations in their interpretation of the protocol or method.

**Reviewer Confidence:**

3: Pretty sure, but there's a chance I missed something. Although I have a good feel for this area in general, I did not carefully check the paper's details, e.g., the math, experimental design, or novelty.

---

> ### Author Rebuttal · Authors · 2023-08-28
>
> Firstly, We'd like to express our appreciation for the comprehensive input regarding our research. We'll proceed to provide clarifications, addressing each of your inquiries and concerns respectively.
> ___
> ### [Response to ‘Questions for the authors’]
>
> **Q1**. From Table 1, the scores of the MCS(dagger) and MCS(star) are almost same. What is the advantage of doing the pre-training? The MCS(dagger) already provides pre-trained models with different languages.
>
> **A1**. The reason we conducted pre-training on the text encoder was to develop our own implementation of the backbone text encoder for use in our experiments. We would like to note that the pre-training methodology itself is not a contribution to our research. Our primary contribution lies in proposing the multilingual CLIPScore (MCS) based on the multilingual CLIP framework, which can be applied to various languages. Therefore, the choice of which multilingual CLIP model to use as a backbone is not a critical issue, as our focus is on introducing MCS that is applicable to multiple languages using the multilingual CLIP foundation. Multilingual CLIP models have been presented not only by Fredrik et al. [1] but also by Others [2]. One reason we chose to utilize our pre-trained text encoder as the backbone for the MCS metric across multiple languages was its superior performance. Subsequently, we conducted experiments on perturbation robustness by further fine-tuning this backbone.
> ___
> **Q2**. For the loss of PR-MCS, the idea is to make the distance between the image embedding and the original caption embedding close and the distance between the perturbed caption embedding and other embeddings further. I wonder if the Triplet loss can be used instead of pair-wise loss. Have you tried or thought about the triplet loss when you design the model?
>
> **A2**. The concept of constructing the loss function through Triplet loss using image embeddings as anchors seems highly insightful. Although we didn't initially contemplate incorporating training through Triplet loss during the model design phase, we've devised a notion for a distinct loss that quantifies distances. This is intended to allow us to assess and fine-tune the contribution of individual loss components using Lambda values. This approach will enable us to measure the influence of each loss and make adjustments accordingly, enhancing the overall effectiveness of our model.
> ___
> **Q3**. In Table 2, the few shot results are based on the different test sets on the regular M-FineCap3k dataset. It can mislead the performance results by reporting under the same dataset. Also, how many few shots are used for the evaluation? It would be more informative if the statistics of the sequence length on the datasets is provided for L510-L516.
>
> **A3**. I agree with your perspective. Since the results in the few-shot setting were obtained using a different test set, it's possible that they could lead to misunderstandings within the same sector of the table. I believe that modifying the table by adding horizontal lines would be helpful to address this issue.
>
> We split M-FineCap3k into subsets proportioned to 1:9 in size and use only 300 perturbed captions as the few-shot input. (L520-523). The statistics for M-FineCap3k, including sequence lengths, can be found in Appendix D and Table 4. The sentences consist of around 20 words each, making them quite lengthy. Following your suggestion, providing this information along with the lengths of MSCOCO, VizWiz, Flickr8k, and Multi30k in lines L510-516 would indeed make the content more informative.
> ___
> **Q4**. In Table 3, PR-MCS results underperform in general except for the Pearson correlation in English. Also, can you provide the significant test results for the correlations? It does not provide convincing evidence to claim that PR-MCS is a metric with a strong correlation with human judgment.
>
> **A4**. In Table 3, a Kendall score of PR-MCS (0.267), similar to CLIPScore's 0.27, is a quite strong indicator of a favorable correlation with human judgment compared to other metrics. Furthermore, the correlation with human judgment measured using Kendall's Tau for other English datasets like Composite and Flickr8k is as follows:
>
> | Metric    | Composite | Flickr8k |
> |-----------|--------------|--------------|
> | BLEU-1    | 0.313        | 0.323	|
> | BLEU-4    | 0.306        | 0.308	|
> | ROUGE-L   | 0.324         | 0.323	|
> | METEOR    | 0.389        | 0.418	|
> | CIDEr     | 0.377         | 0.439	|
> | SPICE     | 0.403        | 0.449	|
> | PR-MCS  | **0.493**	|  **0.527**	|
> | CLIPScore | **0.497**        | **0.530**	 |
>
> The correlation of PR-MCS with human judgment for English sentences is notably strong compared to conventional reference-based metrics and is similar to CLIPScore. Furthermore, as we assert in Table 3, PR-MCS operates in a multilingual setting and demonstrates perturbation robustness, distinguishing it from CLIPScore.
> ___
> ### In conclusion,
> Once again, we appreciate your detailed reviews and insightful comments. We greatly appreciate your insightful input. We've addressed your questions and issues accordingly. Especially considering that you have requested multiple questions regarding the **[Reasons To Reject]**, if our rebuttal has adequately and appropriately addressed those questions, it is our belief that your level of interest and preference towards this study could potentially increase. If these resolutions adequately meet your concerns, we would be grateful if you could consider accepting our paper. In case there are any unresolved matters, please keep us informed.
>
> [1] Manuel Romero Fredrik Carlsson, Ariel Ekgren. 2022. https://github.com/freddefrallan/multilingual-clip.
> [2] Nils Reimers and Iryna Gurevych. 2019. Sentence-bert: Sentence embeddings using siamese bert-networks. arXiv preprint arXiv:1908.10084.

---

### Official Review · Reviewer_J7jR · 2023-08-03

**Typos Grammar Style And Presentation Improvements:** None as far as I know
**Soundness:** 3

**Excitement:**

3: Ambivalent: It has merits (e.g., it reports state-of-the-art results, the idea is nice), but there are key weaknesses (e.g., it describes incremental work), and it can significantly benefit from another round of revision. However, I won't object to accepting it if my co-reviewers champion it.

**Missing References:**

None as far as I know

**Paper Topic And Main Contributions:**

This paper presents PR-MCS, a multilingual metric evaluating image captioning with images and outputs. PR-MCS is built upon multilingual CLIP and further finetuned with three losses with pre-defined perturbation signals. To evaluate the proposed method, the authors in addition propose a multilingual image captioning dataset M-FineCap3k. Experimental results on five languages show that PR-MCS has good performance across dataset sets and languages, and can assign lower scores for the perturbated cases.

**Questions For The Authors:**

Q1: line 199: why the translation process can ensure that the good and bad scores are maintained?

Please refer to Reasons to Reject for more questions.

**Reasons To Accept:**

1.	The authors bring image captioning metric to a multilingual perspective with fair performance. In addition, M-FineCap3K is a useful resource for the community.

2.	The motivation of the work is clear, the reference-free metrics are indeed not robust to lexical perturbations.

**Reasons To Reject:**

1.	The current method seems to have limited contribution: multilinguality of CLIP is from the original multilingual CLIP model itself. The perturbation signals are also inherited from previous work. Conducting further analysis over the selected perturbation type may be good, e.g., showing examples of which the fine-tuning helps, developing a pattern-agnostic method, or if there are further patterns identified. If the model is already aware of which perturbation would be performed, is there any comparison between using the current learned model and using rule-based systems to remove the perturbation?

2.	Lack of comparison with other baseline methods. MCS and PR-MCS are the only metrics compared in the main results table (Table 2). However, other methods such as UMIC (Lee et. al., 2022) can also be included here with minimum adaptation. These different backbone metrics can also be used to further validate the extensibility of the current method.
For the dataset side, authors only compare Multi30K after line 318. What about the Comparison between M-FineCap3k and MSCOCO or VizWiz? Why translation with MBART-50 is not enough? More quantitative or qualitative results needed to be shown.

3.	The results are not clear and sometimes confusing. For example, the correlation between humans and MCS is slight in Table 1: 0.236 on FR does not seem really usable. It would be good to provide some results from reference-based metrics as a comparison. Also, in Table 2, what is the expected scale of the scores? Can the scores be compared across rows?

4.	The dataset is a good resource for the community. However, more verification of the dataset is necessary (e.g., annotation details, annotator agreement, quality assurance methods).

5.	There is slight relation between the multilingual part and the perturbation-prevention part of the paper. I expect more coherence in the proposed contribution. In the area of WSD, various methods have been proposed to leverage multilinguality to improve word sense understanding. Similar approaches can be applied here. For example, Kang et. al., 2023. Translate to Disambiguate: Zero-shot Multilingual Word Sense Disambiguation with Pretrained Language Models.

**Reproducibility:**

3: Could reproduce the results with some difficulty. The settings of parameters are underspecified or subjectively determined; the training/evaluation data are not widely available.

**Reviewer Confidence:**

3: Pretty sure, but there's a chance I missed something. Although I have a good feel for this area in general, I did not carefully check the paper's details, e.g., the math, experimental design, or novelty.

---

> ### Author Rebuttal · Authors · 2023-08-28
>
> We wish to extend our gratitude for the exhaustive insights you have offered in relation to our research. With your thoughtful review in mind, we'd now like to address a few points to provide further context and clarification.
> ___
> ### [Response to ‘Reasons to Reject’]
> **Comment 1**. If the model is already aware of which perturbation would be performed, is there any comparison between using the current learned model and using rule-based systems to remove the perturbation? Conducting further analysis over the selected perturbation type may be good.
>
> **Response 1**. We agree with your viewpoint that assuming the metric's backbone model is aware of the type of perturbation, employing a rule-based approach to remove perturbations can be considered as a method to impart perturbation robustness. However, the process of the model recognizing the perturbation type might also be dependent on parameters. In this study, we opted for a parameter-based approach where the backbone model's parameters inherently recognize perturbations and assign lower scores accordingly. We conducted comparative experiments against the baseline using this method in the current research.
>
> We also find your opinion of "Conducting further analysis over the selected perturbation type may be good" to be very agreeable and insightful. **Figure 1** shows a case study showcasing our fine-tuning technique's ability to achieve perturbation robustness presented for both English and German. We will add more examples, including additional case studies, to the appendix.
> ___
> **Comment 2**. Lack of comparison with other baseline methods. Different backbone metrics can also be used to further validate the extensibility of the current method. For the dataset side, authors only compare Multi30K after line 318. What about the Comparison between M-FineCap3k and MSCOCO or VizWiz? Why translation with MBART-50 is not enough?
>
> **Response 2**. Expanding the experiments to include various backbone models, including UMIC, would indeed add more strength to the analysis in the study. However, CLIPScore demonstrates a higher correlation with human judgment and is more widely used than other reference-free image captioning metrics, including UMIC (Kendall tau 0.530 for CLIPScore, 0.431 for UMIC, and 0.336 for VIFIDEL[1] in case of Flickr8k-Expert benchmark). Considering that our proposed PR-MCS also performs well on good sentences similar to CLIPScore (as shown in Table 3), we believe that conducting experiments using CLIPScore as the backbone provides sufficient support for our methodology analysis.
>
> For the dataset side, the reason for comparing the advantages of the M-FineCap3K dataset with Multi30K lies in the fact that both datasets are multilingual. We constructed the M-FineCap3K dataset to facilitate perturbation across multiple languages, as Multi30K comprises only short sentences and is composed exclusively of languages other than English (only French and German). MSCOCO and VizWiz, being English-only datasets, were not included in the comparison for multilingual image captioning datasets. Since they consist solely of English, they require machine translation models for translation into other languages, which cannot be considered human-annotated datasets. Moreover, like Multi30K, these datasets also share the limitation of containing only short sentences.
> ___
> **Comment 3**. The correlation between humans and MCS is slight in Table 1: 0.236 on FR does not seem really usable. It would be good to provide some results from reference-based metrics as a comparison. Also, in Table 2, what is the expected scale of the scores? Can the scores be compared across rows?
>
> **Response 3**. In Table 1, we have provided numerical values in terms of baseline for reference-free metrics for languages other than English. A comparison between reference-based metrics for M-CapEval1K is presented below.
>
> | Metric    | M-CapEval1K (FR) |
> |-----------|--------------|
> | BLEU-1    | 0.223        |
> | BLEU-4    | 0.224        |
> | ROUGE-L   | 0.210         |
> | CIDEr     | 0.239         |
> | MCS  | 0.236		|
> | PR-MCS | 0.236	|
>
> The value of 0.236 might not seem very high, but considering that the CLIPScore for CapEval1k in English is 0.270, we believe it can serve its purpose as a baseline for French. The CapEval1K demonstrates a higher probability distribution of moderately difficult sentences, ranging from 0.6 to 0.8 on a scale of 1 point. Arranging the sequence of these sentences is more challenging than arranging sequences of obviously good and bad sentences. Therefore, compared to benchmarks with a simple binary distribution of low and high scores, CapEval1K exhibits significantly lower human correlation. This contributes to the fact that CapEval1K itself is a challenging benchmark with a lower correlation to human judgments, partly due to its inherent nature of having a distribution that is not strongly aligned with metrics.
>
> Furthermore, the results of measuring human correlation after translating **Composite** and **Flickr8k** into French using a machine translation model are as follows:
>
> | Metric    | Composite (FR) | Flickr8k (FR) |
> |-----------|--------------|--------------|
> | BLEU-1    | 0.302        | 0.314	|
> | BLEU-4    | 0.303        | 0.302	|
> | ROUGE-L   | 0.317         | 0.311	|
> | CIDEr     | 0.376         | 0.435	|
> | MCS  | **0.497**		|  **0.530**	|
> | PR-MCS  | **0.496**	|  **0.527**	|
>
> In the case of benchmarks like Composite and Flickr8k, MCS and PR-MCS achieved high human correlations (close to 0.5 for Composite, over 0.5 for Flickr8k.). Through this, we believe that the results of MCS and PR-MCS for French are quite usable as image captioning metrics.
>
> Regarding the questions in Table 2, the scales for MCS and PR-MCS range from 0 to 2.5 points. In metric evaluation, we think what matters more than the scale is the correlation with human judgment. In Table 2, as the score drop becomes more pronounced between the original output and perturbed output, it can be interpreted as a better detection of perturbations.
> ___
> **Comment 4**. More verification of the dataset is necessary.
>
> **Response 4**. We have constructed the M-FineCap3k dataset through a professional data collection company, and we have translated the M-CapEval1k Benchmark using a specialized translation agency. In line with your opinion, we believe that detailed information about the dataset construction is crucial for the future utilization of the dataset. The **API and instructions** for collecting the dataset can be found in **Figure 10** and **Figure 11** in the Appendix. We will provide more detailed information about the composition of each dataset, including verification aspects such as annotation details, annotator agreement, and quality assurance methods, in the upcoming version.
> ___
> **Comment 5**. There is slight relation between the multilingual part and the perturbation-prevention part of the paper. I expect more coherence in the proposed contribution.
>
> **Response 5**. We believe that leveraging multilinguality in the Perturbation-prevention part is a promising approach that can greatly contribute to the improvement of the methodology and the analysis of the research. Based on the research directions you have suggested for future studies, we think that developing methods to leverage multilinguality will further advance PR-MCS. We appreciate your insightful and valuable suggestions.
> ___
> ### [Response to ‘Questions’]
> Q1: line 199: why the translation process can ensure that the good and bad scores are maintained?
>
> A1. Across benchmarks for image captioning metrics, including CapEval1k, Flickr8k-expert, and Composite, intentional inclusion of poor sentences alongside reference sentences is employed to assign low scores deliberately. This practice is designed to measure human correlation by incorporating poorly constructed sentences. When utilizing machine translation models, deliberately manipulated poor sentences can be corrected to appear normal through the model's translation. For instance, an English sentence deliberately altered for grammar mistakes, receiving a score of 0.4 out of 1 for quality, might be translated into German with correct grammar. To address this concern, we employed native speakers. In the APIs and instructions detailed in Appendix Figures 10 and 11, annotators were directly instructed to translate the poorly constructed sentences while retaining their low quality.
> ___
> ### In Conclusion,
>
> We greatly appreciate your insightful input. Your valuable feedback is highly valued by us. We have taken the necessary steps to respond to your inquiries and address the matters you raised. Should these solutions effectively address your apprehensions, we would sincerely appreciate your consideration in accepting our paper. However, if there are still areas of concern, we kindly request that you inform us so that we can address them appropriately.
>
> [1] Pranava Swaroop Madhyastha, Josiah Wang, and Lucia Specia. 2019. Vifidel: Evaluating the visual fidelity of image descriptions. In Proceedings of the 57th Annual Meeting of the Association for Computational Linguistics, pages 6539–6550.

---

### Official Review · Reviewer_j2CY · 2023-08-05

**Soundness:** 3

**Excitement:**

4: Strong: This paper deepens the understanding of some phenomenon or lowers the barriers to an existing research direction.

**Paper Topic And Main Contributions:**

This study aims to improve evaluation metrics for image caption generation. Recent CLIP-based evaluation metrics are incapable of correctly evaluating captions with perturbations. To learn a robust evaluation metric model, this study performs contrastive learning between images and captions with and without perturbations. As test data, this study collected multilingual captions annotated with details and important elements. Evaluation on this data shows that the proposed metric can robustly evaluate perturbed captions in five languages while maintaining a high correlation with human evaluations.

**Reasons To Accept:**

1. The susceptibility of evaluation indicators to perturbations is a severe problem; therefore, it is an important contribution to propose a method that could address this.
2. The perturbation evaluation dataset collected in this study will be useful and significantly impact future evaluation metrics studies.
3. This study collected data in English and four other languages and evaluated the evaluation metrics in those languages. This contributes to the construction of more general evaluation metrics.

**Reasons To Reject:**

1. It has not been verified whether the perturbed captions really contain errors in content. It is also unclear whether the artificially created perturbations in this study are similar to those output by the model.
2. N-gram based metrics, e.g., CIDEr and METEOR, are considered relatively robust to perturbations, but this study has not made a comparison.

**Reproducibility:**

3: Could reproduce the results with some difficulty. The settings of parameters are underspecified or subjectively determined; the training/evaluation data are not widely available.

**Reviewer Confidence:**

3: Pretty sure, but there's a chance I missed something. Although I have a good feel for this area in general, I did not carefully check the paper's details, e.g., the math, experimental design, or novelty.

---

> ### Author Rebuttal · Authors · 2023-08-28
>
> First of all, thank you for the thorough feedback on our study.
> We will offer explanations, individually addressing the queries and questions you hold.
> ___
> ### [Response to ‘Reasons to Reject’]
> **Comment 1**. It has not been verified whether the perturbed captions really contain errors in content. It is also unclear whether the artificially created perturbations in this study are similar to those output by the model.
>
> **Answer 1**. We consider your question about the practical utility of perturbation robustness, and we aim to provide responses to this question from two perspectives: rationale and practicality.
>
> Firstly, from a **rationale standpoint**, we want to explain why image captioning metrics should possess perturbation robustness. A good metric should be able to measure good sentences as good and bad sentences as bad. In the foundational work of our study by Sai et al. [1], various criteria in both image captioning and NLG metrics are shown to lack robustness against simple perturbations. These perturbed bad sentences should be evaluated as bad according to these multiple criteria. The perturbations we employed were not directly determined by us empirically, but rather, they were leveraged from the perturbation types outlined by Sai et al. We present methodologies to address these perturbation types.
>
> Moving on to the **practicality perspective**, we have identified errors in the model output for various cases in image captioning. Particularly, issues of "Repetition", where certain tokens are repeated, and "Substitution," where attribute placement is incorrect, occur more frequently in the model's output than other perturbations. These sentences are difficult for reference-free metrics to detect. Although such sentences are not very common, they must be filtered out as bad sentences. Furthermore, the proposed PR-MCS, while maintaining strong human correlation for normal sentences, is a comprehensive metric that maintains perturbation robustness. Thus, it can be practically utilized as it evaluates good sentences as good and has the ability to evaluate bad sentences as bad, unlike existing metrics like CLIPScore.
> ___
> **Comment 2**. N-gram based metrics, e.g., CIDEr and METEOR, are considered relatively robust to perturbations, but this study has not made a comparison.
>
> **Answer 2**. N-gram-based image captioning metrics like BLEU, CIDEr, and METEOR have certain limitations: (1) they require references for sentence evaluation, and (2) their correlation with human judgment is low. The correlation with human judgment for the Flickr8k-expert dataset, as measured using Kendall's Tau, is as follows:
>
> | Metric    | τ_c |
> |-----------|--------------|
> | BLEU-1    | 0.323         |
> | BLEU-4    | 0.308         |
> | ROUGE-L   | 0.323         |
> | METEOR    | 0.418         |
> | CIDEr     | 0.439         |
> | SPICE     | 0.449         |
> | CLIPScore | **0.530**         |
> (With the exception of CLIPScore, all the others are reference-based metrics.)
>
> As a result, recently, reference-free metrics like CLIPScore have been proposed. These metrics do not require references yet exhibit a high correlation score with human evaluations. When it comes to evaluating image captioning models in practical terms, reference-free metrics are more favorable, and among them, CLIPScore is widely adopted.
>
> As per your suggestion, we measured the perturbation robustness of N-gram based metrics for further analysis. Below are the results of the perturbation experiments for N-gram based metrics. Due to time constraints, we did not conduct experiments in a multilingual context. Instead, we examined how well N-gram based metrics capture the presence of perturbations in English sentences. As anticipated, N-gram based metrics demonstrate robustness against perturbations.
>
> |Dataset| Jumble | Removal  | Repetition | Masking | Substitution |
> |-------|-------|-------|-------|-------|-------|
> | MSCOCO | B1:0.99 B4:0.03 R:0.44 C:2.78 S:0.39| B1:0.51 B4:0.27 R:0.69 C:3.30 S:0.42| B1:0.71 B4:0.45 R:0.85 C:4.38 S:0.74| B1:0.60 B4:0.27 R:0.59 C:2.28 S:0.23| B1:0.99 B4:0.29 R:0.67 C:3.25 S:0.45|
> | Flickr8k | B1:0.99 B4:0.03 R:0.40 C:2.77 S:0.37| B1:0.51 B4:0.27 R:0.64 C:3.12 S:0.42| B1:0.71 B4:0.46 R:0.77 C:4.12 S:0.72| B1:0.59 B4:0.27 R:0.53 C:2.11 S:0.21| B1:0.99 B4:0.36 R:0.73 C:4.27 S:0.46|
> | VizWiz| B1:0.99 B4:0.03 R:0.42 C:2.73 S:0.35| B1:0.51 B4:0.26 R:0.70 C:3.02 S:0.41| B1:0.71 B4:0.46 R:0.86 C:4.00 S:0.76| B1:0.59 B4:0.27 R:0.59 C:2.11 S:0.23| B1:0.99 B4:0.39 R:0.71 C:3.75 S:0.39|
>
> (B1, B4, R, C, and S stand for BLEU-1, BLEU-4, ROUGE-L, CIDEr, and SPICE metrics, respectively. The original scores are as follows: B1: 1.0, B4: 1.0, R: 1.0, C: 10.0, S: 1.0.)
>
> Indeed, while N-gram-based metrics demonstrate robustness to perturbations, as mentioned earlier, their value as metrics is considered low due to the limited correlation with human judgment within the current research field. Therefore, we highlight that we have achieved perturbation robustness in reference-free metrics. **We consider high human correlation to be a practical priority when metrics are actually utilized**. Therefore, we have proposed methodologies to further enhance perturbation robustness for CLIPScore.
>
> In summary, the advantages of PR-MCS can be summarized and visualized in the table below:
>
> | Metric | Perturbation Robustness | Correlation with Human Judgment|
> |-------|-------|-------|
> | Reference-based| **Robust(√)** | Low(↓)|
> | CLIPScore | Not robust(x) | **High(↑)**|
> | PR-MCS |  **Robust(√)**  |**High(↑)**|
>
> Certainly, adding this information to the appendix would indeed be helpful in explaining our motivation.
> ___
> ### In Conclusion,
>
> Once again, We’re thankful for your in-depth review and insights. We have provided responses to your concerns and questions. If your concerns are sufficiently addressed, we appreciate it if you would consider accepting our paper. Otherwise, please let us know about your remaining concerns.
>
> [1] Perturbation CheckLists for Evaluating NLG Evaluation Metrics (https://aclanthology.org/2021.emnlp-main.575) (Sai et al., EMNLP 2021)

---

### Meta-Review · Area_Chair_6mRK · 2023-09-17

**Recommendation:** 4

**Metareview:**

This paper introduces a new metric that's robust to perturbations and that can be used for multilingual image captioning. Alongside the metric, a data set is introduced that is used to verify the robustness of this metric.

Pros:
- Tackling an important problem; many other evaluation metrics are highly susceptible to perturbations.
- Multilingual approach, much better than just focusing on one language.
- Introduces both a new metric and a new data set that can both be used by the community.
- Maintaining a strong correlation with human judgment while improving robustness to perturbations.

Cons:
- The comparison to other metrics -- especially as it relates to human judgment correlation -- in the current review copy could be clarified further, e.g. with a table as shown at the bottom of https://openreview.net/forum?id=EpBNf4Arod&noteId=SIuJnYRUqN (it would make sense for this to be in the main body, not relegated to an appendix).
- There's not a lot of detail around how the data set was constructed.

---

### Decision · Program_Chairs · 2023-10-07

**Decision:**

Accept-Findings

**Comment:**

This paper introduces a new metric that's robust to perturbations and that can be used for multilingual image captioning. Alongside the metric, a data set is introduced that is used to verify the robustness of this metric.

Pros:
- Tackling an important problem; many other evaluation metrics are highly susceptible to perturbations.
- Multilingual approach, much better than just focusing on one language.
- Introduces both a new metric and a new data set that can both be used by the community.
- Maintaining a strong correlation with human judgment while improving robustness to perturbations.

Cons:
- The comparison to other metrics -- especially as it relates to human judgment correlation -- in the current review copy could be clarified further, e.g. with a table as shown at the bottom of https://openreview.net/forum?id=EpBNf4Arod&noteId=SIuJnYRUqN (it would make sense for this to be in the main body, not relegated to an appendix).
- There's not a lot of detail around how the data set was constructed.